# Reconciled rat and human metabolic networks for comparative toxicogenomics and biomarker predictions

Edik M. Blais[1], Kristopher D. Rawls[1], Bonnie V. Dougherty[1], Zhuo I. Li[1], Glynis L. Kolling[2], Ping Ye[3,†], Anders Wallqvist[3] & Jason A. Papin[1]

The laboratory rat has been used as a surrogate to study human biology for more than a century. Here we present the first genome-scale network reconstruction of *Rattus norvegicus* metabolism, *iRno*, and a significantly improved reconstruction of human metabolism, *iHsa*. These curated models comprehensively capture metabolic features known to distinguish rats from humans including vitamin C and bile acid synthesis pathways. After reconciling network differences between *iRno* and *iHsa*, we integrate toxicogenomics data from rat and human hepatocytes, to generate biomarker predictions in response to 76 drugs. We validate comparative predictions for xanthine derivatives with new experimental data and literature-based evidence delineating metabolite biomarkers unique to humans. Our results provide mechanistic insights into species-specific metabolism and facilitate the selection of biomarkers consistent with rat and human biology. These models can serve as powerful computational platforms for contextualizing experimental data and making functional predictions for clinical and basic science applications.

[1] Department of Biomedical Engineering, University of Virginia, Box 800759, Health System, Charlottesville, Virginia 22908, USA. [2] Division of Infectious Diseases and International Health, Department of Medicine, University of Virginia, Charlottesville, Virginia 22908, USA. [3] Department of Defense Biotechnology High Performance Computing Software Applications Institute, Telemedicine and Advanced Technology Research Center, US Army Medical Research and Materiel Command, Fort Detrick, Maryland 21702, USA. † Present address: Department of Molecular and Experimental Medicine, Avera Cancer Institute, Sioux Falls, South Dakota 57105, USA. Correspondence and requests for materials should be addressed to J.A.P. (email: papin@virginia.edu).

Rats serve an important role as model organisms in preclinical drug development and biomarker discovery. Candidate drugs are routinely tested in rats to assess safety and efficacy before human clinical trials. Rodent animal models also provide a preclinical platform for characterizing cellular responses to investigational compounds through toxicogenomics analyses of high-throughput molecular data sets[1,2]. Metabolomics profiling of rat serum and urine has been used to quantify potential metabolic biomarkers of drug activity or side effects seen in drug-induced liver injury models[3,4]. Despite a high degree of genomic and physiologic similarities between rats and humans[5,6], functional differences within non-pharmacokinetic metabolism have been described, which could influence whether a compound induces toxicity or elevates a biomarker[7–9]. Understanding species-specific differences between rats and humans will be important for the interpretation of preclinical animal studies in drug development, biomarker discovery and comparative toxicogenomics analyses[2,10,11].

A genome-scale network reconstruction (GENRE) of metabolism acts as a repository for all known biochemical and transport reactions for an organism. Several GENREs with thousands of human genes have been published[12–15], while only core metabolic networks with dozens of genes are available for rat[16,17]. A high-quality GENRE of rat metabolism is needed to bridge the knowledge gap that exists between humans and rats in clinical and basic science applications. The limited availability of highly curated GENREs for rats and other animals has been attributed to the substantial efforts required to manually construct a GENRE based on information from biochemical databases, genome annotations and literature evidence[18].

A comprehensive collection of metabolic differences between rats and humans would be a valuable resource for understanding the applicability, as well as the limitations of rats in preclinical drug development and biomarker discovery[10,11]. Human GENREs have been used to predict metabolic biomarkers for inborn errors of metabolism (IEMs)[13,19] and to analyse the metabolic effects of therapeutic strategies in the context of cancers, toxicology and diabetes[14,15,20,21]. Computational methods for integrating gene and protein expression measurements into GENREs have been developed to generate context-specific metabolic networks and enable comparative predictions across individual patients, treatment conditions and tissue types[15,22–24]. Resolving metabolic differences between rat and human GENREs would enable cross-species comparisons as previously described for bacterial GENREs[25,26]. However, the lack of unified standards for metabolic networks[27] has limited the development of computational frameworks for analysing species-specific differences between GENREs.

In this study, we construct the first comprehensive GENRE of rat metabolism and a newly updated GENRE of human metabolism. We manually curate both rat and human metabolic networks in parallel to reconcile species-specific differences and facilitate cross-species comparisons. As a result, these models successfully capture known metabolic features that distinguish humans and rats. To demonstrate the use of these models in systems toxicology, we integrate high-throughput transcriptomics data to predict biomarker changes in response to 76 environmental and pharmaceutical compounds for rat and human hepatocytes. Comparisons of rat and human biomarker predictions provide mechanistic insights into a human-specific side effect caused by theophylline distinct from that of the structurally related compound, caffeine. We validate select biomarker predictions for these two xanthine derivatives with literature-based evidence and new experimental data. Overall, the comparative network analyses between rat and human metabolism presented here provide a novel framework for improving the translation of future preclinical studies in rats to humans.

## Results

**Overview of rat and human metabolic networks.** GENREs of *Rattus norvegicus* (iRno) and *Homo sapiens* (iHsa) metabolism were constructed in parallel as an expansion of the Human Metabolic Reaction 2.0 database[15] (HMR2). To enable comparative systems analyses, we created a unified reaction database termed Ratcon1 that includes the superset of metabolic and transport reactions that occur in rats and humans. Each GENRE also includes gene protein reaction (GPR) rules to describe genotype to phenotype relationships that are organism specific. To establish an initial draft GENRE of rat metabolism, we used orthology information to replace human GPR rules with rat GPR rules (Fig. 1a). Next, we updated iRno and iHsa in parallel with 169 new reactions, 1,103 manually reconciled GPR relationship rules and over 5,000 additional references to experimental literature and annotation databases[28–30] (Fig. 1b and Supplementary Data 1). Compared with previous human and mouse GENREs[12,13,15,31,32], iRno and iHsa captured the highest numbers of total reactions, enzymatic reactions, reactions associated with complex GPR rules and annotations to external databases (Table 1). As previous rat metabolic networks were created for the purpose of metabolic flux analysis within the scope of central metabolism, to our knowledge iRno represents the first genome-scale network model of rat metabolism. Furthermore, all reactions were reconciled for potential differences between rat and human networks, which has not previously been described for existing mammalian networks (Table 1). As a result, iRno and iHsa represent two of the most comprehensive metabolic reconstructions and the first pair of mammalian metabolic networks reconciled for comparative analyses to date.

**Network reconstruction and reconciliation.** Before manual curation, we assembled a draft of iRno based on a draft of iHsa using orthology annotations from five online resources: the Rat Genome Database[33], Homologene[34], Ensembl[35], the Kyoto Encyclopedia of Genes and Genomes[28,29] and the Universal Protein Resource (UniProt)[36] (Fig. 1a). Inferring function through orthology can be difficult[37] when individual human genes are annotated to multiple rat orthologues (Supplementary Figs 1 and 2). To identify a minimal subset of orthologues that preserved basic functionalities in the draft of iRno, we assigned confidence to 4,768 orthologue pairs between 2,588 human genes and 2,897 rat orthologues using a consensus approach (see Methods). As a result, we selected a subset of 2,629 orthologue pairs between 2,499 human genes and 2,575 rat orthologues that were annotated in at least two databases (Supplementary Data 2), to generate a draft of iRno that was iteratively improved in parallel with iHsa (Fig. 1b). Without filtering orthology data, the draft rat network was difficult to compare with the original human network due to differences in the numbers of redundant enzymes associated with each reaction (see Supplementary Fig. 3).

After network reconciliation, there was a high degree of confidence in the conserved metabolic functionality of iRno and iHsa. Unexpectedly, we found that rat and human metabolic networks were distinguished by as few as eight unique enzymes. At the genome scale, 99.6% of all gene-associated reactions were annotated with both rat and human genes (Supplementary Data 3). We simulated the effects of species-specific differences on network connectivity and found that 41 biochemical or transport reactions were uniquely capable of carrying flux in

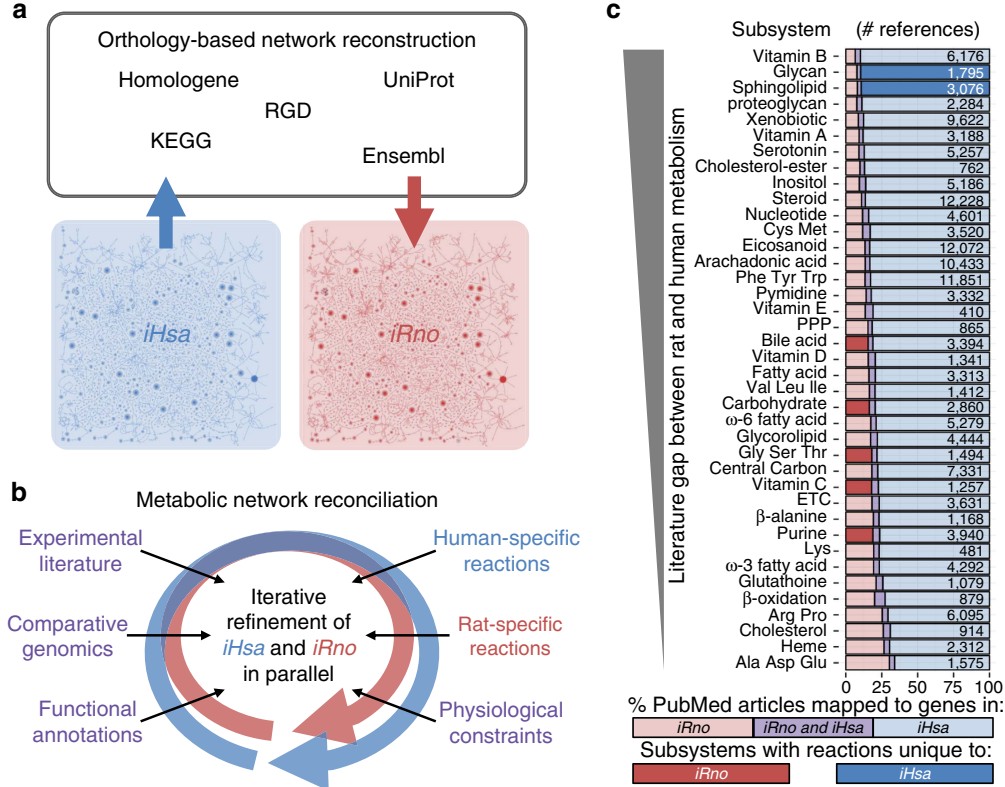

**Figure 1 | GENREs of human (*iHsa*) and rat (*iRno*) metabolism were reconciled for comparative analyses.** (**a**) Reactions from *iHsa* were transferred to *iRno* when GPR rules consisting of human genes could be replaced with equivalent GPR rules consisting of rat orthologs. A network-driven approach was developed to filter orthology annotations based on consensus from five databases: Homologene, KEGG, Uniprot, Rat Genome Database (RGD) and Ensembl. (**b**) *iRno* and *iHsa* were manually curated in parallel to capture species-specific reactions and to avoid introducing unverifiable differences based on genotype and phenotype information across various resources. This process, called network reconciliation, contributed to the identification and removal of several rat-specific reactions that are present in HMR2 and Recon 2. (**c**) Subsystem-level comparison of the knowledge gap that exists between rats and humans. Stacked bars represent the percentages of PubMed articles mapped to rat and/or human genes for all metabolic genes represented in a subsystem. PubMed articles referenced human genes more frequently than rat genes within all subsystems, but the knowledge gap was larger for pathways that included one or more human-specific reactions. ETC, electron transport chain; PPP, pentose phosphate pathway.

**Table 1 | Comparison of reconciled rat and human GENREs with previous mammalian GENREs.**

| Database | Ratcon1 | | HMA | | BiGG | | VMH |
|---|---|---|---|---|---|---|---|
| **Organism**<br>**Model** | Rat<br>*iRno* | Human<br>*iHsa* | Mouse<br>MMR | Human<br>HMR2 | Mouse<br>*iMM1415* | Human<br>Recon 1 | Human<br>Recon 2 |
| Genes* | **2,324** | **2,315** | **3,579** | **3,728** | **1,375** | **1,496** | **1,733** |
| Reactions | **8,268** | **8,263** | **8,140** | **8,181** | **3,726** | **3,742** | **7,440** |
| Enzymatic* | 5,745 | 5,738 | 5,597 | 5,604 | 2,204 | 2,297 | 4,446 |
| Isozymic* | 2,863 | 2,691 | 3,013 | 3,135 | 776 | 832 | 1,647 |
| Enzyme complex* | 620 | 620 | 0 | 0 | 237 | 250 | 461 |
| Annotated in KEGG | 2,412 | 2,406 | — | 1,527 | — | — | — |
| Species-specific | 14 | 7 | 18 | 60 | 0 | 100 | — |
| Unreconciled† | 0 | 0 | 62 | 85 | 83 | 17 | — |
| Average GPR size | 2.98 | 2.97 | 3.66 | 3.89 | 1.83 | 1.91 | 1.97 |
| Metabolites | **5,620** | **5,620** | **5,516** | **5,546** | **2,775** | **2,766** | **5,063** |
| Unique metabolites | 3,200 | 3,200 | 3,170 | 3,155 | 1,503 | 1,509 | 2,626 |
| Compartments | 8 | 8 | 8 | 8 | 8 | 8 | 8 |
| Biomass metabolites | 184 | 169 | 117 | 117 | 41 | 41 | 41 |
| Annotated in KEGG | 3,353 | 3,353 | — | 689 | — | — | 2,601 |
| Metabolic tasks | **327** | **327** | **56** | **256** | **254** | **260** | **354** |
| Species-specific‡ | 12 | 2 | — | — | — | — | — |

GENRE, genome-scale network reconstruction; GPR, gene protein reaction; HMA, Human Metabolic Atlas; BiGG, Biochemical Genetic and Genomic knowledgebase of large scale metabolic reconstructions; VMH, Virtual Metabolic Human; KEGG, Kyoto Encyclopedia of Genes and Genomes.
*iRno* and *iHsa* expand on HMR2 with curated GPR rules that account for enzyme complexes, updated annotations to external databases, and no unreconciled† differences between species-specific models.
*Reactions associated with highly redundant GPR rules (10 + isozymes) such as generic signalling processes were excluded in these model comparisons.
†Unlike species-specific reactions, which are enzymatic in one species and absent in the other, unreconciled reactions can either be enzymatic in one species and non-enzymatic in the other, or non-enzymatic in one species and absent in another.
‡Species-specific tasks are explicitly designed to succeed in one species and not the other.

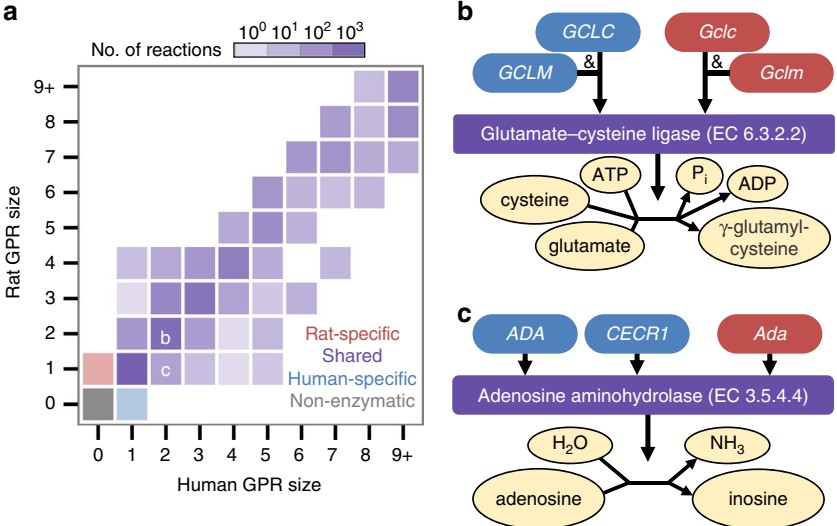

**Figure 2 | Reconciled GPR relationships between *iRno* and *iHsa* allow for varying degrees of redundancy. (a)** Comparison of the number of genes catalysing each reaction in *iRno* and *iHsa*. Gene-associated reactions capable of catalysis in both *iRno* and *iHsa* were classified as 'shared' reactions (purple). Reactions associated with GPR rules in only one organism were classified as species-specific (red and blue for rat-specific and human-specific, respectively). Reactions present in both models that had no known GPR rule assignments were classified as non-enzymatic (grey). Each tile's colour intensity represents the (log-scaled) frequency of reactions in that bin. The annotated letters **b,c** refer to the individual tiles from which the reactions in **b,c** are binned. (**b**) Example of a shared reaction with balanced GPR rules in *iRno* and *iHsa*. This reaction, glutamate-cysteine ligase (EC 6.3.2.2), requires both a catalytic subunit (*Gclc/GCLC*) and a regulatory subunit (*Gclm/GCLM*) to join glutamine with cysteine and form γ-glutamyl-cysteine, a precursor of glutathione. This reaction was manually curated, because the original HMR2 did not contain information related to protein complexes in GPR rules. (**c**) Example of a shared reaction, adenosine aminohydrolase (EC 3.5.4.4), which is involved in purine degradation and can be catalysed by two redundant human isozymes or one rat enzyme.

either *iRno* or *iHsa*. This result was unanticipated, because 739 flux carrying reactions from HMR2 had been disabled in the draft GENRE of rat metabolism. Despite extensive efforts to identify metabolic activities unique to rat or human genomes (see Supplementary Methods), most metabolic subsystems included zero species-specific reactions after manual curation (Fig. 1c).

To approximate whether sufficient literature information was available to identify known species-specific differences, we compared how frequently PubMed articles referenced rat and human genes within individual subsystems. We found that rat genes were referenced less frequently compared with human genes, although the literature gap between rat and human genes varied substantially by subsystem (Fig. 1c). Subsystems with human-specific reactions included fewer references to rat genes relative to human genes compared with other subsystems. Interestingly, the number of reactions classified as human specific based on orthology annotations decreased from 19 to 7 after performing network reconciliation (Supplementary Data 1), suggesting a higher degree of consistency between rat and human metabolic capabilities than currently annotated. Alternatively, rat-specific reactions were not identified for any of the poorly studied subsystems in rats (Fig. 1c), suggesting that additional studies may reveal undiscovered differences between rat and human metabolism.

Metabolic enzymes unique to either rats or humans more frequently contributed to increased redundancies rather than new functionalities when comparing the relative sizes of rat and human GPR rules across shared reactions (Fig. 2). GPR sizes were consistent (along the diagonal of Fig. 2a) for nearly 80% of reactions associated with both rat and human genes (an example is illustrated in Fig. 2b); however, known differences in the numbers of redundant rat and human genes have been described[38] for reactions such as the example shown in Fig. 2c. Capturing variability between rat and human GPR rules is

important, because the numbers of redundant isozymes or subunits in a protein complex affect the relative robustness of reactions to genetic perturbations. Despite individual variations in GPR sizes and a handful of species-specific reactions, rat and human GPR rules remained relatively balanced at the genome scale (Fig. 2a) and were not suggestive of any global differences in robustness within metabolism. However, these rat and human GPR formulations do not reflect potential tissue-specific differences in gene expression or enzyme regulation. Integration of such data into *iHsa* and *iRno* to generate tissue-specific models could be of significant interest in numerous biological contexts[13–15,22,23].

**Metabolic tasks captured known species-specific functions**. We assembled a comprehensive collection of 327 metabolic tasks that captured known functions within rat and/or human metabolism. Each task represented a known biological process such as producing glucose from lactate during gluconeogenesis or breaking down glutamine into $CO_2$ and urea. As a result, we recapitulated 14 new species-specific tasks, 42 new shared tasks and 271 shared tasks previously described in the validation of human metabolic network reconstructions[13,15,20] (Supplementary Data 4). Species-specific tasks were well represented across multiple subsystems including ascorbate, purine, glycan and bile acid metabolism (Fig. 1c), and each task was characterized by one or two unique enzymatic reactions (Fig. 3). Below, we showcase the importance of capturing these differences with *iRno* and *iHsa* in the contexts of human biology and disease.

Unlike humans, rats are capable of producing vitamin C (ascorbate; Fig. 3a) and are thus resistant to scurvy[39]. *iRno* and *iHsa* captured this species-specific phenomenon with a new task that simulated *de novo* vitamin C synthesis in a glucose minimal media environment (Supplementary Data 4). The rat-specific

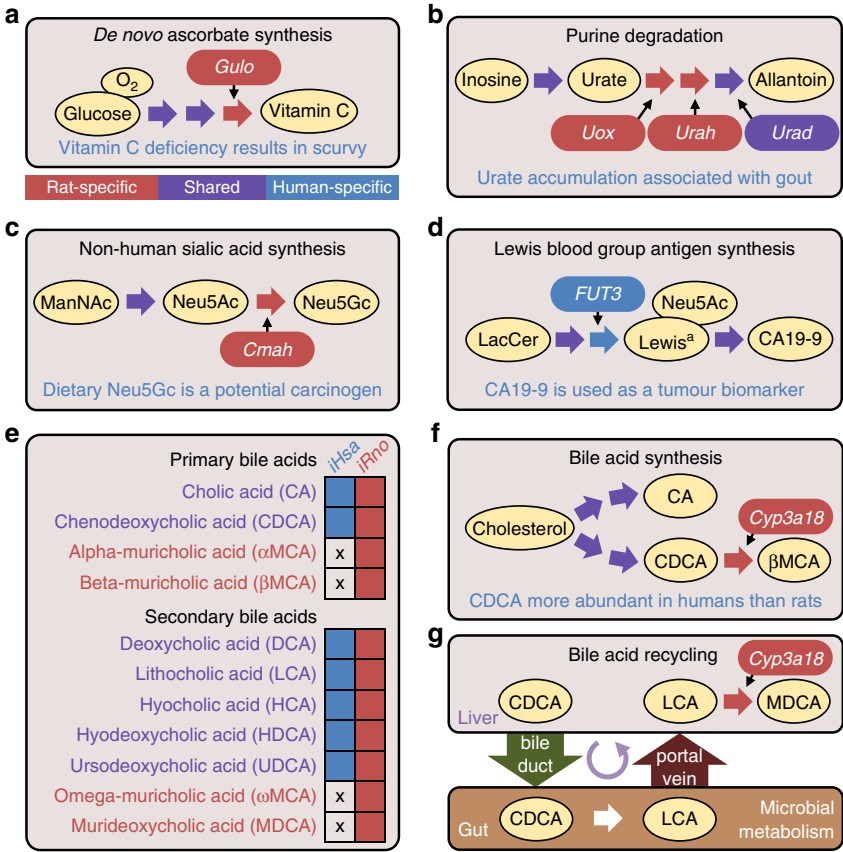

**Figure 3 | Functional differences known to distinguish rat and human metabolism.** (**a**) Rats are capable of synthesizing vitamin C from limited substrates, providing an inherent resistance to scurvy. The rat-specific enzyme, *Gulo*, catalyses the last enzymatic step of the vitamin C synthesis pathway: L-gulonolactone oxidase (EC 1.1.3.8). The human orthologue of *Gulo* is a non-functional pseudogene (*GULOP*). (**b**) Gout formation is associated with accumulation and crystallization of urate, the end product of purine catabolism in humans. Rats are resistant to gout, because urate can be further degraded into allantoin, which is more soluble than urate. (**c**) Most mammals can synthesize the monosaccharide, Neu5Gc, which is known as a nonhuman sialic acid that is incorporated into glycoproteins. (**d**) The human-specific enzyme *FUT3* synthesizes the Lewis[a] antigen (Le[a]) which is involved in Lewis blood group determination and is the precursor for the pancreatic cancer biomarker, CA19-9. (**e**) Metabolic tasks simulating the production of primary and secondary bile acids were consistent with bile acid species reported in a previous study[48] that compared rat and human liver samples (X's indicate absence of production). (**f**) Summary of rat-specific and shared primary bile acid synthesis routes from cholesterol. *Cyp3a18* was hypothesized as the critical enzyme enabling rats to produce rat-specific primary bile acids, which had not been previously described. (**g**) In the process of 'bile acid recycling', primary bile acids secreted by the liver into the gut are transformed by bacteria and subsequently reabsorbed by the liver. Synthesis of secondary bile acids were accounted for in *iRno* and *iHsa* by including extracellular reactions associated with gut bacteria. Interplay between rat liver and gut metabolism was necessary for *iRno* to simulate the synthesis and secretion of the rat-specific bile acid, murideoxycholic acid (MDCA). CA19-9, carbohydrate antigen 19-9; LacCer, lactosylceramide; ManNAc, *N*-acetylmannosamine; Neu5Ac, *N*-acetylneuraminic acid; Neu5Gc, *N*-glycolylneuraminic acid.

enzyme, *Gulo*, is known to be responsible for this functional difference, which limited the use of rat as a model organism for scurvy in the early twentieth century[40]. This species-specific difference provides a simplistic example of how *iRno* and *iHsa* can be leveraged to investigate genetic engineering strategies that bridge the gap between rat and human biology (see Supplementary Fig. 4).

In humans, the purine degradation pathway yields urate as the end byproduct, which can accumulate and cause gout[41]. Rats can further degrade urate into allantoin (Fig. 3b) and are resistant to gout formation[42]. *iRno* and *iHsa* captured this functional difference with new metabolic tasks that simulated the production of urate from purines, which is common to both species, and allantoin from purines, which is absent in humans (Supplementary Data 4). The first two steps involved in converting urate into allantoin, urate oxidase (EC 1.7.3.3) and 5-hydroxy-isourate hydrolase (EC 3.5.2.17), are catalysed by the rat-specific enzymes *Uox* and *Urah*, respectively. The human orthologues for

these two rat-specific genes, *UOXP* and *URAHP*, are non-functional pseudogenes; however, the third and last enzymatic step, 2-oxo-4-hydroxy-4-carboxy-5-ureidoimidazoline decarboxylase (EC 4.1.1.97) is a shared reaction encoded by *Urad* in rats and *URAD* in humans.

**Known differences in glycosylation.** Most mammals, including rats, can synthesize *N*-glycolylneuraminic acid, a sialic acid found in glycolipids and glycoproteins[43,44], via the enzyme *Cmah*, cytidine monophosphate-*N*-acetylneuraminic acid hydroxylase (EC 1.14.18.2) (Fig. 3c). Humans cannot produce *N*-glycolylneuraminic acid from *N*-acetylneuraminic acid, a prevalent sialic acid in humans, because the orthologue *CMAHP* is a non-functional pseudogene in humans[44]. Despite this difference, human sialyltransferases can incorporate non-human sialic acids into glycans obtained through the consumption of red meat[43], which we also captured as a shared task in rats and humans (Supplementary Data 4).

The human-specific enzyme, *FUT3*, encodes a fucosyltransferase involved in the Lewis blood group system. An individual with a functional copy of *FUT3* can produce the Lewis a antigen[45] (Le[a]) and sialyl-Le[a], a clinical biomarker for pancreatic cancer commonly referred to as carbohydrate antigen 19-9 (Fig. 3d). Despite the inability of rats to synthesize Le[a], we expect that carbohydrate antigen 19-9 could be produced from exogenous Le[a] by the orthologous sialyltransferases *St3gal3* and *ST3GAL3* (ref. 46). Surprisingly, *FUT3* was the only functional difference attributed to human-specific enzymes after performing network reconciliation between *iRno* and *iHsa*.

**Bile acid pathway curation and reconciliation**. The bile acid metabolic pathway was expanded in *iRno* and *iHsa* to include bile acids that may serve as biomarkers in rats and humans[47] (Fig. 3e). Human hepatocytes can synthesize chenodeoxycholic acid and cholic acid from cholesterol, to facilitate dietary lipid absorption (Fig. 3f). In addition to these two primary bile acids, rat hepatocytes also produce large quantities of α-muricholic and β-muricholic acid that are absent in humans[47,48] (Fig. 3f). After extensive manual curation of this pathway (described below), metabolic tasks simulating the synthesis of four primary and seven secondary bile acids were consistent with previous experiments directly comparing bile acids detected in the livers and sera of rats and humans[47,48] (Fig. 3e and Supplementary Data 4).

The mammalian intestinal microbiome plays an important role in converting primary bile acids synthesized by hepatocytes into secondary bile acids, which can be re-absorbed and further metabolized in the liver[49]. This 'bile acid recycling' expands the global pool of metabolites encountered by humans and rats beyond what their individual genomes allow. To account for bile acid recycling in *iRno* and *iHsa*, we introduced a new extracellular subsystem of 'gut' metabolic reactions that converted primary bile acids into secondary bile acids (Fig. 3g). This simplified system representing the intestinal microbiome was necessary to capture the synthesis of murideoxycholic acid, a rat-specific bile acid derived from the secondary bile acid, lithocholic acid (Fig. 3g and Supplementary Data 4).

Although curating the bile acid synthesis pathway, we discovered that the critical enzymatic step involved in the production of rodent-specific bile acids was not annotated to any rat or mouse genes (Fig. 3e). A previous study hypothesized that an unknown cytochrome P450 family 3 member could produce β-muricholic acid and murideoxycholic acid via 6-β hydroxylation of chenodeoxycholic acid and lithocholic acid, respectively[50]. Using Basic Local Alignment Search Tool (http://www.uniprot.org/blast/), we compared the Golden Hamster (*Mesocricetus auratus*) gene *Cyp3a10* (UniProt ID: Q64148), which has been reported to perform 6-β hydroxylation of lithocholic acid[51] (EC 1.14.13.94), with rat genes. We identified *Cyp3a18* as the best candidate with the highest sequence identity to *Cyp3a10* (Supplementary Data 5) and with protein-level evidence of expression in rat hepatocytes[52]. Furthermore, *Cyp3a18* was the only potential match with no known human orthologues, consistent with the absence of this function in humans[48]. In contrast, a traditional Basic Local Alignment Search Tool comparison of *Cyp3a18* against other mammalian genomes resulted in three mouse genes with higher sequence identity but different functional annotations compared with *Cyp3a10*, highlighting how reconciling metabolic network reconstructions can guide the improvement of genome annotations[53].

**Metabolic network improvements**. Network reconciliation efforts provided significant improvements to both *iRno* and *iHsa*. Although most of the species-specific functions described above

are unique to rats, we discovered that previous human GENREs included rat-specific reactions associated with purine degradation, non-human sialic acid synthesis and glycine metabolism. As a result, rat-specific reactions were not only added to *iRno* but also removed from *iHsa*. By resolving species-specific differences in the purine degradation pathway, we removed reactions from *iRno* and *iHsa* that allowed Recon 2 (ref. 13) and HMR2 (ref. 15) to degrade urate into urea (Supplementary Data 1), a function known to be absent in mammals but present in other vertebrates including fish[42]. Although curating bile acid metabolism, we removed intracellular reactions involved in secondary bile acid synthesis (Fig. 3f) that should only take place in the mammalian gut and added new bile acid transport reactions (Supplementary Data 1). These examples highlight how reconciling differences between rat and human metabolism guided the improvement of *iHsa* compared with HMR2 and Recon 2.

The manual curation process introduced major differences between *iHsa* and previous human GENREs beyond species-specific pathways. The average GPR size across enzymatic reactions decreased from 3.89 in HMR2 to 2.97 in *iHsa* (Table 1) after removing 1,424 human genes, most of which were associated with signalling pathways (Supplementary Data 1). With an average GPR size of 1.97 (Table 1), Recon 2 shared 1,531 of its 1,733 genes with *iHsa*. Although Recon 2 shared 1,677 genes with HMR2, *iHsa* and Recon 2 included complex GPR relationships that were absent in HMR2 (Table 1). In addition to modifying GPR rules, we also removed several reactions that were present in HMR2 (Supplementary Data 1). Unlike Recon 1, Recon 2 and HMR2, *iHsa* does not include thermodynamically infeasible reaction loops that drive unrealistic rates of ATP production and $H_2O_2$ detoxification with limited nutrients (see Supplementary Methods). We formulated new shared metabolic tasks that captured physiologically relevant ATP yields from glucose with and without oxygen, and verified that no ATP could be 're-phosphorylated' with only inorganic ions as inputs (see Supplementary Data 4).

**Benchmarking biomarker predictions for IEMs**. Metabolic biomarkers are routinely screened to pinpoint genetic deficiencies in metabolic enzymes and to diagnose IEMs[19]. We evaluated the ability of *iHsa* to predict known metabolic biomarkers for 49 IEMs (Supplementary Fig. 5). Metabolites were predicted as elevated, reduced or unchanged for *iHsa*, HMR2 and Recon 2 using data previously described in the validation of Recon 2 (ref. 13). *iHsa* correctly predicted 83% of 99 IEM biomarkers compared with 81% for HMR2 and 82% for the most recent iteration of Recon 2 (Table 2). For IEM predictions, we applied open constraints to exchange reactions that were more consistent across *iHsa*, HMR2 and Recon 2 than default constraints (see Methods). Compared with predictions described in the original Recon 2 publication[13] (Table 2), predictions for all three human GENREs were slightly more sensitive for elevated biomarkers but less sensitive for reduced biomarkers. We also explored possible

**Table 2 | Sensitivity of *iHsa* in predicting known biomarkers of IEMs compared with previous human reconstructions.**

| IEM biomarker | Count | *iHsa* | HMR 2.0 | Recon 2.04 | Recon 2.00 |
|---|---|---|---|---|---|
| Elevated | 83 | 80 | 78 | 77 | 66 |
| Reduced | 16 | 3 | 2 | 4 | 10 |
| Total | 99 | 83.8% | 80.8% | 82.8% | 76.8% |

IEM, inborn errors of metabolism.

species-specific differences and found that most GPR rules associated with IEM mutations were closely mirrored by equivalent rat GPR rules.

**Physiological constraints for hepatocyte growth.** To interrogate the use of these reconstructions for making cell-specific predictions, we defined quantitative biomass compositions for rat and human hepatocytes (Supplementary Fig. 6 and also see Supplementary Methods). Using flux balance analysis[54] (FBA) with biomass production as the objective, *iRno* and *iHsa* predicted maximum growth rates of 0.048 and 0.040 h$^{-1}$, respectively, under strict physiological constraints (Supplementary Fig. 7). These predicted doubling times of 14.44 h by *iRno* and 17.33 h by *iHsa* were remarkably consistent with reported doubling times of 16.9 h in rat[55] and 17.8 h in human[56] hepatocyte cell cultures. As biomass compositions and boundary conditions were independently formulated from different resources, these quantitative biomass predictions served as validation for these models and their comprehensive representations of hepatocellular growth (Table 1). Physiological constraints also enable off-the-shelf use of *iRno* and *iHsa* for integration of comparative genomics data and systems-level analyses of hepatocyte metabolism.

**Systems toxicology applications of *iRno* and *iHsa*.** Rats are often used as a surrogate model for understanding human hepatotoxicity; consequently, it is critically important to understand species-specific responses to experimental compounds, to efficiently translate preclinical studies. To explore the effects of exposure to pharmaceutical compounds and environmental toxicants on normal metabolic functions, high-throughput gene expression profiles of rat and human hepatocytes were obtained from the Open Toxicogenomics Project-Genomics Assisted Toxicity Evaluation system[2,10] (Open TG-GATEs) and analysed within the computational frameworks of *iRno* and *iHsa*. We preprocessed raw microarray data from the Open TG-GATEs independently for 119 individual compounds and calculated gene expression changes between control samples and samples treated with a low, medium or high dose for 8 h (see Methods). Of 119 compounds with expression data available, 76 were considered suitable for comparative toxicogenomics analyses after excluding treatments that did not significantly affect (false discovery rate (FDR) < 0.1) at least 1% of the 1,927 or 2,175 metabolic genes that mapped to *iRno* or *iHsa*, respectively (Supplementary Data 6).

To demonstrate the utility of *iRno* and *iHsa* in biomarker discovery for human toxicology, we generated biomarker predictions for rat and human hepatocytes exposed to these 76 environmental toxicants and pharmaceuticals (Supplementary Data 7). Species-specific gene expression changes in response to 76 compounds were integrated into *iRno* and *iHsa* using transcriptionally inferred metabolic biomarker response (TIMBR; Fig. 4), a new algorithm that estimates the feasibility of producing a metabolite given changes in gene expression (see Methods). First, TIMBR summarizes gene expression log$_2$ fold changes into reaction weights that represent the relative cost or demand of carrying flux through each reaction for treatment and control conditions (Fig. 4a,b). Second, TIMBR calculates the global network demand required for biomarker production by minimizing the weighted sum of fluxes across all reactions for each condition (Fig. 4c,d). This general approach, known as parsimonious enzyme usage FBA[57], was previously adapted for integrating absolute gene expression measurements (present or absent)[24,58] but not for relative gene expression changes (upregulated or downregulated) as done here with TIMBR.

By integrating relative changes in gene expression, TIMBR predictions represented the relative propensity to produce

metabolites in response to an individual compound. As a result, relative production scores were determined independently for each treatment by normalizing TIMBR predictions across all exchangeable metabolites (Fig. 4e). By applying similar physiological constraints to *iRno* and *iHsa* (Supplementary Fig. 7) and requiring similar production rates for each metabolite (Fig. 4b), TIMBR provided a novel framework for making biomarker predictions across metabolites, treatments and organisms. In contrast, a similar approach that integrated absolute expression was described as capable of making comparisons across experimental conditions but not between individual metabolites[58]. As most species-specific reactions take place in peripheral pathways (Fig. 3), it is not immediately apparent that there would be species-specific differences in the ability to produce a metabolic biomarker under physiological constraints. However, TIMBR biomarker predictions can account for species-specific differences in gene expression patterns with an explicit mapping of species-specific GPR rules as accounted for in *iRno* and *iHsa* (Fig. 2).

**Caffeine-induced biomarker predictions for rat hepatocytes.** We validated TIMBR as a quantitative method for predicting relative metabolic changes in response to caffeine using previously published data[59] (Fig. 5). Caffeine-induced gene expression changes from rat hepatocytes (Fig. 4b) were integrated into *iRno* using TIMBR to generate biomarker production scores (Fig. 4e). An increased production score for a metabolite such as urea indicated that genes involved in urea synthesis and secretion were more consistently upregulated than downregulated by caffeine. Reaction weights and fluxes that contributed to urea production in caffeine treatment and control conditions are visualized in Fig. 4d. We quantitatively compared TIMBR production scores (Fig. 5) with previously reported concentration changes after caffeine exposure in rats[59]. We evaluated our biomarker predictions against serum levels of urea and ten additional metabolites measured in rat liver samples[59]. We found that rat production scores based on *in vitro* gene expression data significantly correlated (Pearson's $r = 0.667$; $P$-value = 0.0249) with caffeine-induced liver metabolic changes reported *in vivo* (Fig. 5). In addition, all metabolites that were experimentally elevated (urea, citrulline and aspartate) or reduced (glutamate) by caffeine treatment were consistently predicted in the top or bottom 25% of production scores, respectively. For metabolites that were not significantly affected after caffeine treatment, most TIMBR predictions were within the middle 50% of production scores, with the exception of ornithine and arginine. The TIMBR algorithm also successfully predicted *in vivo* metabolite concentration changes in response to caffeine with a Matthew's correlation coefficient of 0.69, indicating both high sensitivity (100%) and specificity (71%). As TIMBR predictions are based on transcriptional changes and do not rely on any knowledge of a compound's mechanism of action, we anticipate this computational approach will be broadly applicable to any compound that induces a detectable physiological response.

**Comparative biomarker predictions.** Species-specific differences in the metabolic response to a drug candidate could hamper the successful translation of preclinical biomarkers of efficacy or toxicity from rats to humans. We compared TIMBR predictions generated by integrating gene expression changes into *iRno* and *iHsa*, and found a weak but significant positive correlation (Pearson's $r = 0.1958$; $P$-value < 10$^{-11}$) between rat and human production scores across 286 metabolites and 76 compounds. We analysed rat and human production scores predicted in response to individual compounds and categorized 40 as positively correlated,

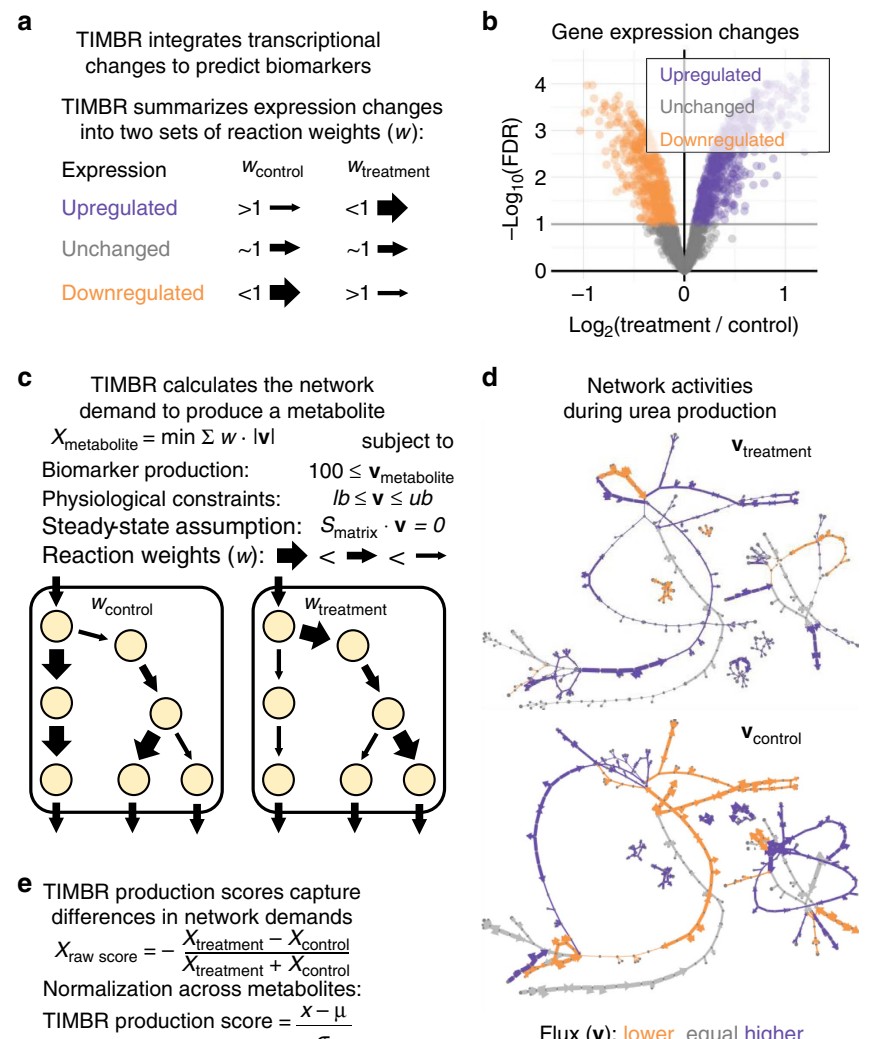

**Figure 4 | TIMBR is a novel method for predicting treatment-induced biomarkers by integrating gene expression changes into metabolic networks.**
(**a**) TIMBR calculates reaction weights using $\log_2$ fold changes of significantly (FDR < 0.1) differentially expressed genes. For each reaction, $\log_2$ fold changes are averaged across isozymes after assigning a value of 0 to any insignificant changes. For reactions associated with protein complexes, the subunit with the largest value after averaging is selected. Summarized values are then transformed into larger (or smaller) reaction weights for representing relative expression between treatment and control conditions. (**b**) Caffeine-induced gene expression changes are displayed as a volcano plot for rat hepatocytes. (**c**) Optimization problem formulated by TIMBR to estimate the global network demand needed to produce a metabolite. The objective function minimizes the sum of all reaction fluxes (**v**) multiplied by TIMBR reaction weights (*w*). Treatment and control conditions were simulated separately for each potential biomarker under similar physiological lower-bound (*lb*) and upper-bound (*ub*) constraints that assumed steady-state reaction fluxes. The minimum required production rate for each metabolite was set to either a rate of 100 fmol per cell per hour or 90% of the maximum possible flux value, whichever was smaller. (**d**) Optimal caffeine-weighted ($w_{treatment}$) and control-weighted ($w_{control}$) flux distributions ($\mathbf{v}_{treatment}$ and $\mathbf{v}_{control}$) for biomarker production of urea determined by integrating gene expression changes from (**b**) into *iRno*. Non-zero fluxes that were higher (purple), equal (grey) or lower (orange) relative to the other condition were displayed using MetDraw[70]. Arrow thickness represents the inverse reaction weight as described in **a**. In this example, the global network demand (sum of weighted fluxes) was smaller in the treatment condition than the control condition, indicating that caffeine induced expression changes that were more consistent with the production of urea compared to controls. (**e**) Raw production scores in response to individual treatment strategies were calculated for each metabolite separately by comparing global network demands determined in **c** for the treatment and control conditions. TIMBR production scores represent these raw production scores normalized across all relevant metabolites with biomarker predictions.

23 as uncorrelated and 13 as negatively correlated using an FDR significance threshold of 0.1 (Supplementary Data 7).

We validated TIMBR predictions against known metabolic changes related to the therapeutic efficacy for antipyretic and anti-gout medicines. Ibuprofen and acetaminophen are over-the-counter antipyretics that are known to inhibit cyclooxygenase enzymes (COX-1 and COX-2 encoded by *PTGS1* and *PTGS2*)[60]. We compared rat and human biomarker predictions for individual metabolites across all 76 compounds and found that rat and

human production of prostaglandin E2, a metabolite synthesized downstream of COX1/2, was predicted to decrease in response to acetaminophen and ibuprofen (Fig. 6b). For anti-gout compounds, we analysed the predicted effects of benzbromarone[61], benziodarone[61], colchicine[62] and phenylbutazone[63] on urate production. Despite differences in chemistry, we found that both rat and human production scores for urate were decreased for three out of four anti-gout medications, consistent with their abilities to decrease urate accumulation (Fig. 6c). Furthermore,

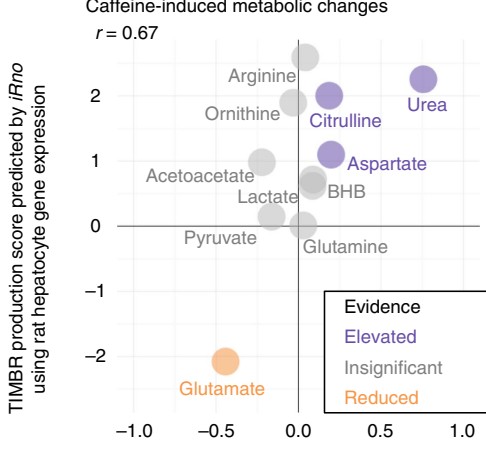

**Figure 5 | Validation of caffeine-induced biomarker predictions for rat hepatocytes.** Comparison of rat production scores calculated by TIMBR in response to caffeine with previously reported average log$_2$ fold changes in metabolite concentrations after caffeine exposure[59].

urate production was predicted to increase in response to several compounds for human hepatocytes but rarely for rat hepatocytes (Fig. 6c), consistent with known species-specific differences in purine degradation (Fig. 3b).

**Validation of species-specific biomarker predictions**. TIMBR biomarker predictions were generally consistent between *iRno* and *iHsa* for caffeine but not for theophylline, despite the fact that both compounds are structurally related derivatives of xanthine (Fig. 6). We investigated species-specific differences and found that theophylline-induced urate production was predicted to increase for human hepatocytes and decrease for rat hepatocytes (Fig. 6c). As validation, we experimentally confirmed that extra-cellular urate levels were decreased for primary rat hepatocytes and increased for a human hepatocyte cell line treated with theophylline for 24 h (Fig. 7). In contrast, caffeine was predicted to decrease urate production in both rat and human hepatocytes (Fig. 6c), despite known differences in the urate degradation pathway (Fig. 3b). By comparing reaction weights and fluxes associated with urate production, we found that shared reactions involved in purine synthesis and purine degradation were uniquely upregulated in human hepatocytes by theophylline and not by caffeine (Supplementary Fig. 8). Interestingly, caffeinated beverages have been associated with a decreased incidence of hyperuricemia in patients[64] (Fig. 6c), whereas theophylline has been reported to increase serum urate levels in patients (hyperuricemia)[65]. Using only toxicogenomics gene expression data as an input into *iRno* and *iHsa*, TIMBR provided comparative predictions that led to mechanistic insights into how two nearly indistinguishable compounds (Fig. 6f) induced similar responses in rat (Fig. 6d) but drastically different responses in human (Fig. 6e)[66].

To support the use of the TIMBR algorithm in comparative toxicogenomics analyses across species, metabolites and compounds, we experimentally measured extracellular concentrations of urate (as described above), glutamate, glucose and urea after treating rat and human hepatocytes with 0 or 10 μM theophylline for 24 h (Fig. 7). As a result, we confirmed predicted species-specific differences in urate (Fig. 7a), glutamate (Fig. 7b) and glucose levels (Fig. 7c), and supported a shared trend towards increased urea production (Fig. 7d). Individually, seven out of eight experimental results were qualitatively consistent with significantly (FDR < 0.05) elevated, reduced or insignificant concentration changes (Fig. 7). Similar to our analyses described for caffeine (Fig. 5), we found that theophylline-induced changes in metabolite concentrations were quantitatively consistent with TIMBR production scores generated by *iRno* (Pearson's $r = 0.959$; $P$-value $= 0.041$) and *iHsa* (Pearson's $r = 0.963$; $P$-value $= 0.034$) (Fig. 7e). In addition, our validated rat predictions related to glutamate (Fig. 7a) and urea (Fig. 7d) were also consistent with our validated predictions for caffeine (Fig. 5), supporting analyses between treatments as highlighted in Fig. 6d. Overall, these results demonstrate the utility of *iRno* and *iHsa* in integrating gene expression data, generating functional predictions and in identifying species-specific biomarkers.

## Discussion

This study provides a systems-level overview of species-specific differences between rat and human metabolism captured with GENREs. Through the process of network reconciliation[25], we discovered that rats and humans share an overwhelming majority of their biochemical capabilities at the genome level, underscoring the important role of rats as a model organism for understanding human biology and disease. To demonstrate the use of these highly curated rat and human metabolic networks in systems toxicology, we developed a novel platform for comparative toxicogenomics analyses and integrating high-throughput genomics data sets. These mechanistic models of cellular metabolism faithfully recapitulated known species-specific metabolic functions, quantitatively captured cellular growth rates and generated comparative biomarker predictions.

With an improved understanding of rat and human metabolism at a global level, we can partially address inherent limitations in the use of rats to study human physiology and disease. Despite a surprisingly small number of species-specific differences at the genome level, individual differences at the gene-level can alter network functionality. Unlike rats, humans exclusively rely on dietary sources of vitamin C, which may obfuscate the clinical translation of rat studies that have described vitamin C as a potential biomarker[7,9]. The abundance and absence of β-muricholic acids in rats and humans, respectively, can have substantial implications within the context of toxicology, because bile acids are frequently used as blood-based biomarkers of liver damage[8]. Furthermore, species-specific differences in gene network regulation and downstream cellular responses to stimuli have been observed but are not well understood[67].

The comparative toxicogenomics workflow developed in this study could be used to further the translational impact of rats in biomarker discovery by highlighting metabolic biomarkers that should be avoided. Using relative changes in gene expression, we demonstrated the sensitivity of the TIMBR algorithm in predicting species-specific differences related to glutamate, glucose and urate production in response to theophylline. With the ability to analyse biomarker predictions between treatments and across metabolites, TIMBR could also be informative in prioritizing biomarkers that are sensitive for a specific toxicological response.

Together, *iRno* and *iHsa* serve as a computational resource for the understanding of rat metabolism within the context of human biology. *iRno* and *iHsa* have the capacity to improve the effectiveness of rat as a model organism in drug development and biomarker discovery. Potential applications include identifying combinatorial therapeutic strategies against cancers that minimize toxicity in normal cells[21], optimizing cell culture media

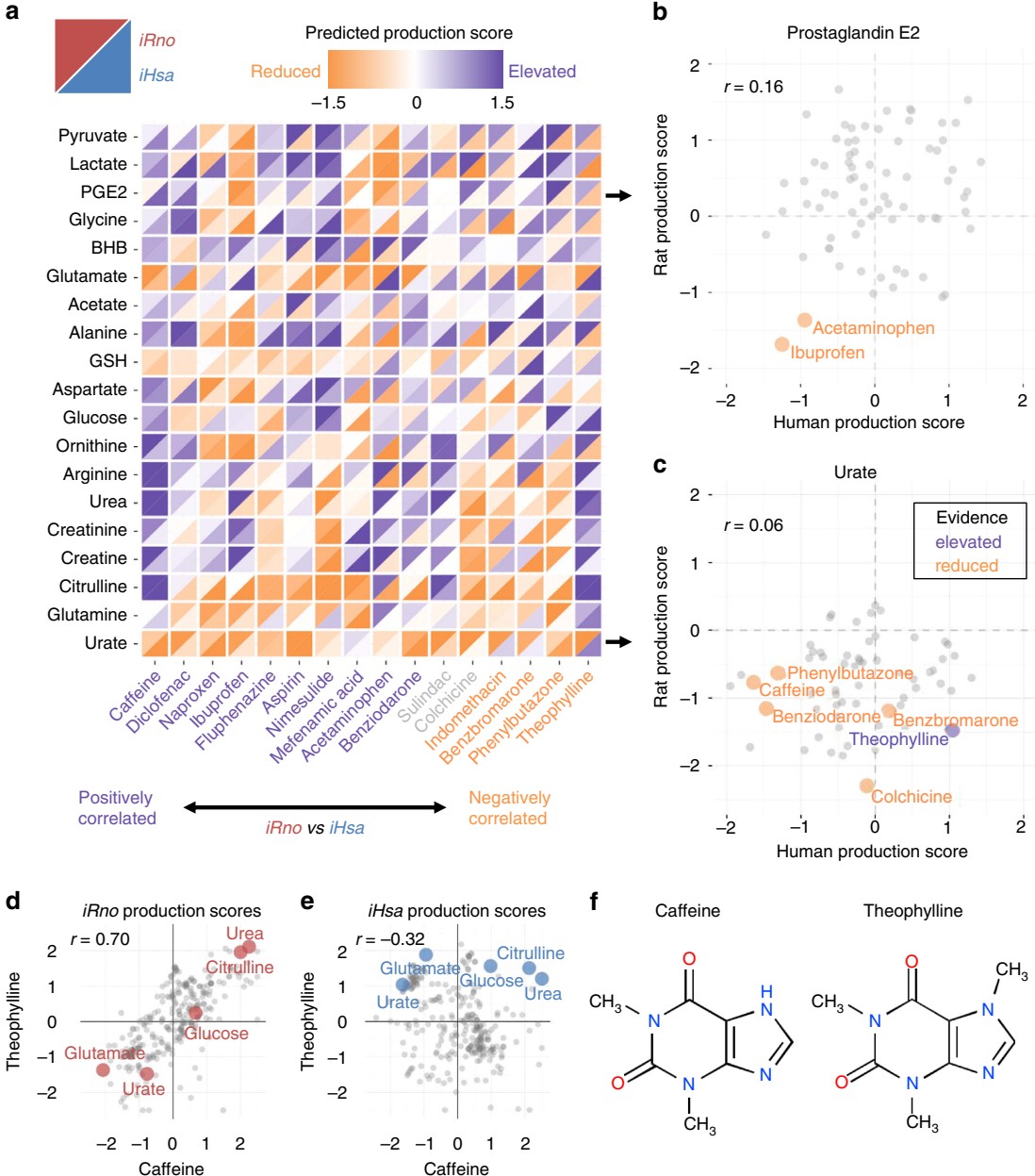

**Figure 6 | Comparative analyses of rat and human biomarker predictions and species-specific trends across perturbations. (a)** Heatmap of 16 metabolite biomarkers predicted to increase (purple) or decrease (orange) in response to 16 individual compounds. Metabolite production scores for rat (upper left triangle) and human (lower right triangle) hepatocytes were generated by integrating treatment-induced gene expression changes into *iRno* and *iHsa* using TIMBR. Rat and human production scores across all 286 metabolites were classified as positively correlated (FDR < 0.1), uncorrelated or negatively correlated (FDR < 0.1) for each individual compound. Compounds were ordered by correlation coefficients and metabolites were ordered by average production scores across all 76 compounds. BHB, β-hydroxybutryate; PGE2, prostaglandin E2. **(b)** Scatterplot comparing rat and human production scores for PGE2 across 76 compounds. Two antipyretic compounds with known cyclooxygenase inhibitor activities, acetaminophen and ibuprofen, were predicted to consistently decrease prostaglandin E2 production in both rat and human hepatocytes. **(c)** Scatterplot comparing rat and human production scores for urate across 76 compounds. Rat production scores for urate were consistently decreased by anti-gout medications that are known to reduce urate accumulation (colchicine, phenylbutazone, benziodarone and benzbromarone). Human production scores were also decreased for anti-gout compounds with the exception of benzbromarone. **(d)** Rat production scores in response to two xanthine derivatives, caffeine and theophylline, were strongly correlated. Biomarker predictions associated with glutamate, urate, glucose and urea were individually consistent across both compounds. **(e)** Human production scores in response to theophylline and caffeine were less correlated than rat production scores. Glutamate and urate were predicted to increase in response to theophylline and decrease in response to caffeine for human hepatocytes, whereas glucose and urea predictions were consistent across both compounds. Compared with rat production scores (**d**), urate, glucose and glutamate would be considered species-specific predictions. **(f)** Chemical structures for theophylline and caffeine differ by a single methyl group. Rat biomarker predictions did not indicate any potential differences between the two xanthine derivatives. In patients, theophylline is known to cause increased serum levels of urate[65].

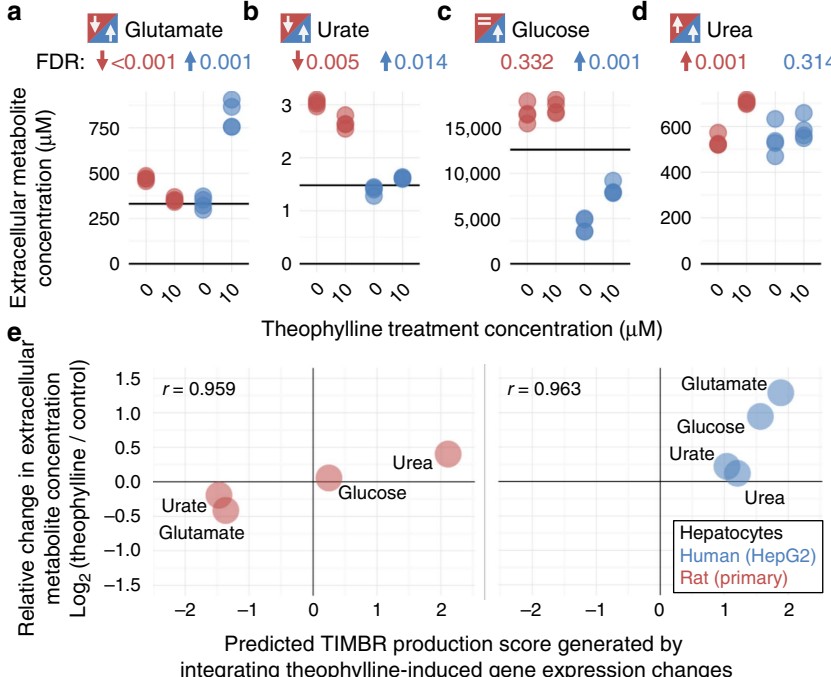

**Figure 7 | Validation of *iRno* and *iHsa* biomarker predictions in response to theophylline.** (**a–d**) Comparative biomarker predictions were experimentally validated *in vitro* using primary rat hepatocytes or an immortalized human hepatocyte cell line (HepG2). Extracellular changes in metabolite concentrations were measured after 24 h of treatment with theophylline (10 µM) or control (0 µM). To the left of each metabolite, predictions are summarized as elevated (up arrow), reduced (down arrow) or unchanged (equals sign). Below each measured metabolite, FDR-adjusted q-values are displayed for rat and human experimental comparisons between treatment ($n = 4$) and control ($n = 4$) sample concentrations using an unpaired two-sided Student's t-test. Individual points represent biological replicates from one rat and one human experiment. Of the eight predictions tested, seven were experimentally confirmed, while urea from HepG2 cells (**d**) was insignificant but trending in the expected direction. Horizontal lines represent average metabolite concentrations from two fresh media replicates and indicate that all metabolites were being produced on average with the exception of glucose in HepG2 cells. (**e**) Quantitative comparisons between model predictions and experimental results across metabolites. Positive and negative values represent elevated and reduced biomarkers based on TIMBR production scores (x axis) and average log₂ fold changes between treatment and control (y axis) concentrations displayed in **a–d**.

formulations or experimental diets for specific diseases and exploring potential genetic engineering strategies for new rat strains that better mimic human biology. Resources can be prioritized for simulated experiments that minimize species-specific differences *in silico* over those not supported by the mechanistic framework of these reconciled rat and human metabolic networks.

## Methods

**Draft reconstruction of a rat metabolic network.** Several published metabolic network reconstructions were considered for the basis of *iHsa* and *iRno*. These included HMR2 (ref. 15), *H. sapiens* Recon 1 (ref. 12) and Recon 2 (ref. 13), HumanCyc[30] and Hepatonet1 (ref. 20); ultimately, the largest network, HMR2, was chosen for its inclusivity. In addition, HMR2 was capable of performing 256 well-curated metabolic tasks relevant to hepatocyte metabolism in both humans and rats. Constructing *iRno* based on a mouse metabolic network was also considered; however, each mouse model was derived from one of the human models described above[31,32].

To construct a rat metabolic network from a human metabolic network, human GPR rules were converted into rat GPR rules by replacing human genes with known rat orthologues (Fig. 1a). Pairs of orthologous genes between humans and rats were downloaded from five separate genomics databases: the Rat Genome Database[33], Ensembl[35], Kyoto Encyclopedia of Genes and Genomes[28,29], Uniprot[36] and Homologene[34]. Each individual database contained varying amounts of orthology information and unique orthology pairs, suggesting a lack of consensus (see Supplementary Methods and Supplementary Fig. 2). All five databases were used in the GPR conversion process, because inferring function through orthology can be especially difficult when considering genes with multiple orthologues (Supplementary Fig. 1). Before GPR conversion, annotations mapping a human gene to ten or more rat orthologues were excluded to avoid inferring function from

nonspecific annotations. A draft of *iHsa* was adapted from HMR2 by replacing human GPR rules consisting of Ensembl gene identifiers with protein-coding Entrez genes. Ensembl genes without equivalent protein-coding Entrez genes were discarded before GPR conversion. In addition, several reactions with excessively large GPR rules such as generic protein kinase reactions were simplified or discarded to streamline the GPR conversion process (Supplementary Data 1).

Aggregating orthology annotations from multiple databases increases the risk of inappropriately replacing human genes with rat orthologues that do not perform the same function. A consensus approach was used to filter out low-quality orthology annotations during the GPR conversion process. Each orthologous pair of rat and human genes was assigned a score of 1–5 corresponding to the number of databases in which that pair was annotated. Individual genes were also assigned confidence scores (Supplementary Data 2) determined by protein-level evidences and annotation scores from Ensembl[35] and Uniprot[36] (see Supplementary Methods). For human genes mapped to multiple rat genes, orthologues were prioritized first by database scores then confidence scores to assign orthology ranks. Sensitivity analysis was performed to explore how filtering out orthologue pairs based on different cutoff values for database scores and orthology ranks affected the distributions of rat and human GPR sizes (see Supplementary Methods). Ultimately, a subset of 2,629 orthologue pairs were selected that were annotated in at least 2 of 5 databases and limited each human gene to a maximum of 2 orthologues (Supplementary Data 2). This filtering step was important, because methods that integrate gene expression data or simulate the impact of genomic alterations rely heavily on the number of redundant enzymes associated with a reaction.

After applying automated methods to construct draft GENREs, rat and human GPR rules were manually examined in the context of experimental literature and functional databases. As part of the network reconciliation process, discrepancies were resolved for both rat and human GPR rules when possible. Manual changes frequently affected both rat and human GPR rules (Supplementary Data 1), highlighting the reliance of this automated approach on the accuracy of GPR rules from the original model and the quality of orthology annotations. Interestingly, the need to filter orthology annotations was only realized after manually inspecting

evidence for both rat and human GPR rules. Converting all orthology annotations present in any of the five orthology databases generated rat GPR rules with disproportionately more genes compared with the original human GPR rules (Supplementary Fig. 3). The consensus-based approach for filtering orthology annotations was designed to minimize the introduction of species-specific differences between rat and human GPR rules (see Supplementary Methods), consistent with the principles of network reconciliation[25]. Furthermore, humans and rats have similarly sized genomes with ∼20,000 genes, so genome-scale properties were assumed to be consistent unless manually curated GPR rules based on literature evidence suggested otherwise. The automated reconstruction methods described in this study could also be used to generate draft GENREs for other organisms by mapping orthology annotations to an existing GENRE. Importantly, this network-driven approach to determine thresholds for orthology annotation filtering can accommodate varying amounts of consensus-based orthology information available for an organism. This novel approach can also be modified to prioritize orthology annotations based on other data types such as protein sequences similarity[36], overlapping functional annotations[29] or homologous gene clusters[34].

During the manual curation process, new reactions were added to iRno and/or iHsa, which were not previously present in HMR2 (Supplementary Data 3). Identification of species-specific reactions was prioritized, because rat-specific reactions were unlikely to be included in a human GENRE (see Supplementary Methodsfor details). Interestingly, some rat-specific functions were already included as non-gene associated reactions in HMR2 and deleted from iHsa as part of the reconciliation process. Furthermore, many reactions originally annotated as human-specific were considered shared reactions after identifying appropriate rat enzymes (Supplementary Data 1). Ultimately, 14 rat-specific and 7 human-specific metabolic reactions were included in iRno and iHsa (Table 1) in addition to 16 artificial reactions involved in the formation of species-specific components used in the biomass formulation (Supplementary Fig. 6).

### Resolving known species-specific differences in metabolism.
Network reconciliation was emphasized throughout the entire reconstruction process for iHsa and iRno to facilitate cross-species predictions. Oberhardt et al.[25] compared GENREs of two closely related Pseudomonas species developed independently and found that cross-species predictions were unrealistic without extensive network reconciliation. The percentage of reactions shared between the two bacterial models increased from 33 to 86%, achieved mostly by resolving differences in nomenclature used to describe reactions and metabolites themselves. Between iRno and iHsa, a much higher percentage (>99%) of reactions was shared, probably as a result of bypassing the need to reconcile terminology-based differences between species. In contrast, the per cent of reactions that overlapped between previous mouse and human GENREs was closer to 98% due in part to reactions that were not reconciled (Table 1).

In addition to resolving differences at the reaction level, iRno and iHsa were further reconciled for comparative analyses by manually updating GPR rules. A major disadvantage of using HMR2 for the basis of iRno and iHsa was the absence of complex GPR relationship rules with multiple subunits in a protein complex (Table 1). Manually curated GPR rules containing protein complexes were based primarily on GPR relationships from H. sapiens Recon 2 (ref. 13), experimental literature and genome annotation databases (see Supplementary Data 1). As a result, GPR rules with multiple subunits were constructed for 620 reactions in iRno and iHsa (see Fig. 2b for example). After network reconciliation and extensive manual curation, the numbers of rat and human genes mapped across shared reactions remained balanced (Figs 2a and 3c).

### Biomass formulations.
A novel system of reactions representing biomass synthesis was developed to enable cross-species predictions of growth between iRno and iHsa with a single biomass reaction (Supplementary Fig. 6). Using quantitative values from hepatocyte-based experimental literature (see Supplementary Methods and Supplementary Data 1 for details), iRno and iHsa included reactions that consumed known quantities of individual metabolites to produce an 'average' biomass precursor metabolite that was species independent. For biomass precursors with relatively similar compositions such as the average nucleotide incorporated into DNA (Supplementary Fig. 6b), macromolecular synthesis reactions were shared by iRno and iHsa. Species-specific macromolecular synthesis reactions were added to represent distinct compositions of bile acids (Supplementary Fig. 6c) and amino acids obtained from studies comparing metabolomics profiles of rat and human hepatocytes[47,48,68]. This generalized framework for biomass formulations was implemented in iRno and iHsa to simulate hepatocyte growth and can be extended to formulate species-specific compositions for groups of metabolites in any tissues with quantitative or comparative metabolomics data. In addition, hepatocyte growth and production of each of the eight macromolecular precursors under strict physiological constraints were simulated as separate metabolic tasks (Supplementary Data 4). Strict physiological constraints were defined as inputs (Supplementary Fig. 7) for quantitative predictions of maximum growth rates using the unified biomass reaction as the cellular objective for iRno and iHsa (see Supplementary Methods for additional details). It is important to note that upper- and lower-bound constraints were consistent across all shared rat and human reactions, including physiological constraints for exchange reactions. Thus, species-

specific differences in growth rate predictions could be explained by minor differences in biomass requirements for essential amino acids.

### Simulation of metabolic tasks.
Metabolic tasks representing known biological functions of rats and humans were simulated in iRno and iHsa using the Reconstruction, Analysis and Visualization of Metabolic Networks (RAVEN) toolbox[15,69]. These included 256 metabolic tasks from HMR2 (ref. 15) and 15 tasks adapted from Recon 2 (ref. 13). An additional set of 53 new tasks were defined including 14 species-specific tasks such de novo synthesis of vitamin C (Fig. 3a). Identifying species-specific tasks and tasks related to hepatocyte metabolism was prioritized to capture biological functions that might be important for the use of these models in studying toxicology (see Supplementary Data 4). As a result, nearly all species-specific enzymes were also supported by functionally important species-specific task.

As none of the original 271 metabolic tasks were considered unique to humans, metabolic task simulations were expected to be consistent between iRno and iHsa. Before manual curation, the automated draft of iRno originally failed to complete three human tasks related to bile acid synthesis that have been described as functional in rats[48]. Assigning the rat gene Akr1c14 to 3α-hydroxysteroid dehydrogenase (EC 1.1.1.50) was sufficient to resolve all three inconsistent metabolic task predictions between iRno and iHsa[38]. In contrast, the mouse GENRE, iMM1415, required the addition of 95 reactions to complete 260 metabolic tasks after automated conversion from H. sapiens Recon 1 (refs 12,32 and Table 1). As orthology between Akr1c14 and AKR1C4 had not previously been annotated in any of the five orthology databases (Supplementary Data 2), we manually investigated all reactions that were annotated as human specific after automated reconstruction of iRno to resolve differences attributed to missing orthology annotations.

### Biomarker predictions for IEMs.
The ability to predict known metabolic biomarkers for IEMs was evaluated using iHsa. Known associations between genes and metabolites for various IEMs were evaluated as previously described in the validation of H. sapiens Recon 2 (refs 13,19). Biomarkers change for each IEM were estimated by comparing feasible flux ranges via flux variability analysis for metabolite exchange reactions between healthy and unhealthy conditions. Healthy and unhealthy conditions were simulated by forcing and disabling flux through reactions associated with an IEM, respectively. For the healthy condition, individual reactions associated with IEM genes were constrained to 90% of the maximum possible flux value determined by FBA under open exchange conditions as described previously[13]. Open exchange conditions were formulated to allow unconstrained uptake of 12 inorganic ions (Supplementary Fig. 7d) and limited uptake (−1 arbitrary units) of all metabolites with exchange reactions. Biomarker prediction performance was measured by the sensitivity to detect known biomarkers of IEMs. The performance of iHsa to predict 99 biomarker/IEM pairs was compared with Recon 2 (ref. 13) (version 2.04) and HMR2 (ref. 15).

### Toxicogenomics analysis of rat and human hepatocyte data.
Gene expression profiles of rat and human hepatocytes treated with 119 different compounds were obtained from the Open TG-GATEs (http://toxico.nibiohn.go.jp)[2,10]. Raw microarray data were downloaded from ArrayExpress (E-MTAB-797 for rat hepatocytes; E-MTAB-798 for human hepatocytes) and pre-processed using the oligo package in the R/Bioconductor programming environment. Expression changes after 8 h of treatment were independently determined for each compound and organism using the limma package. Genes with a FDR-corrected q-value <0.1 were considered significantly differentially expressed. Of the 119 compounds with data available for both rat and human hepatocytes, 76 were selected for model integration that significantly altered at least 1% of the 1,925 rat genes or the 2,177 human genes common to both microarrays and models.

### Transcriptionally inferred metabolic biomarker predictions.
Biomarker changes in response to 76 pharmaceutical compounds and environmental toxicants were predicted for rat and human hepatocytes with TIMBR, a novel constraint-based analysis algorithm (Fig. 4). TIMBR integrates gene expression data from treatment and control samples, and calculates a production score for each exchangeable metabolite under relaxed physiological constraints (Supplementary Fig. 7). TIMBR production scores represent the consistency between the reactions needed to synthesize and secrete a potential biomarker and the relative expression of genes associated with those reactions. For each metabolite, a production cost was calculated by minimizing the total weighted flux across all reactions while maintaining positive flux through its extracellular exchange reaction:

$$\text{Minimize } [w \cdot |\mathbf{v}|] \text{ subject to } S \cdot \mathbf{v} = 0; \mathbf{v}_{lb} \leq \mathbf{v} \leq \mathbf{v}_{ub}] \quad (1)$$

In equation (1), $\mathbf{v}$ is a vector of reaction fluxes, $w$ is a scalar vector of reaction weights based on gene expression measurements, $S$ is the stoichiometric matrix, and $v_{lb}$ and $v_{ub}$ are scalar vectors of lower- and upper-bound constraints for reaction fluxes. To simulate physiologically relevant conditions, nutrient uptake was limited to physiological values by setting lower-bound constraints of metabolite exchange reactions to quantitative values derived from experimental literature (Supplementary Fig. 7b). To simulate production of a potential biomarker,

non-zero positive flux was forced by setting the lower-bound through a metabolite's exchange reaction to 90% of the maximum possible secretion rate determined by flux variability analysis under relaxed physiological constraints or a value of $100\,fmol\,cell^{-1}\,hour^{-1}$, whichever was smaller. To solve this optimization problem with a linear programming solver, metabolic networks were first converted into irreversible metabolic networks where each reversible reaction was represented by separate forward and reverse reactions in the stoichiometric matrix and $v_{lb}$ is non-negative.

Reaction weights based on treatment-induced changes in gene expression were calculated independently for each compound in each organism. Given a set of gene expression changes, TIMBR generates two sets of reaction weights representing treatment and control conditions (Fig. 4a). For the optimization problem in equation (1), each set of reaction weights represent the costs associated with carrying flux through reactions in treatment or control conditions (Fig. 4b). In general, reaction weights for the treatment condition are increased by downregulated genes and decreased by upregulated genes, whereas reaction weights for the control condition are decreased by downregulated genes and increased by upregulated genes. In contrast to approaches that integrate gene expression measurements as weights for flux minimization[24,58], TIMBR uses differential expression instead of absolute expression values.

To transform relative gene expression changes into reaction weights for treatment and control conditions, TIMBR implements a novel approach for summarizing multiple expression changes through GPR relationships. In general, GPR rules use Boolean operators to describe multiple genes that encode redundant isozymes with an 'OR' relationship and genes that encode subunits in an enzyme complex with an 'AND' relationship. For a group of isozymes, the $log_2$-fold change was averaged such that the effect of one upregulated isozyme could either be offset by downregulation in another isozyme or diluted by the presence of multiple unaffected isozymes. For subunits in an enzyme complex, the $log_2$-fold change with the largest absolute value was used (see Supplementary Methods for additional details). Expression changes were summarized for $log_2$-fold changes, because the distributions of $log_2$-fold changes were more evenly distributed (Fig. 4b). Summarized reaction values based on $log_2$-fold changes were inverse log transformed and multiplied to the default vector of reaction weights to represent the control condition (Fig. 4a). For the treatment condition, default reaction weights were divided by reaction fold change values such that upregulated reactions contributed less to the sum of weighted fluxes than downregulated reactions and vice versa for controls (Fig. 4a). Default weights of one for biochemical reactions and two for transport reactions were doubled for reactions with no gene associations or expression data available. Treatment and control condition weights were then applied separately to either iRno or iHsa, to calculate the global network demand (sum of weighted fluxes) for the production of each potential biomarker (Fig. 4c).

Production scores representing relative biomarker changes were determined by comparing biomarker production costs based on treatment and control reaction weights (Fig. 4e). Raw production scores were calculated based on the relative global network demand defined in equation (1) between treatment and control conditions for each biomarker:

$$\text{Raw production score} = \frac{\text{Control} - \text{Treatment}}{\text{Treatment} + \text{Control}} \quad (2)$$

Raw production scores from equation (2) across all potential metabolic biomarkers were normalized independently for each compound in each organism using a $z$-score transformation (Fig. 4e). With this method, positive or negative production scores could be interpreted as the increased or decreased propensity for a metabolite to be synthesized and secreted in response to a treatment relative to other metabolites. To determine whether rat and human hepatocytes were more or less similar in their metabolic response to individual compounds, production scores were analysed across all 286 potential biomarkers shared between iRno and iHsa. Biomarker-level similarity was assessed by calculating the correlation coefficient between rat and human production scores across all compounds. Similarly, consistencies were determined for individual compounds by calculating the correlation coefficient between rat and human production scores across all metabolites.

Compounds classified as xanthine derivatives, xanthine oxidase inhibitors and antipyretics were selected for subsequent analysis and visualization. Potential biomarkers were manually chosen to include PGE2 and urate, which are directly downstream of enzymes targeted by COX-2 inhibitors (antipyretics) and xanthine oxidase inhibitors (anti-gout compounds), metabolites with validation data and select common metabolites.

The TIMBR algorithm was developed to provide a proof-of-principle application of iRno and iHsa in comparative toxicogenomics analyses; however, significant challenges remain in generating quantitatively accurate biomarker predictions that can facilitate the clinical translation of preclinical studies in rats with this method. High-throughput validation with untargeted metabolomics data would be necessary to comprehensively assess the predictive ability of the TIMBR algorithm across many subsystems in these metabolic networks. As transcriptional changes do not immediately have an impact on metabolite abundances, accurate knowledge of metabolite kinetics in response to a perturbation might be important when designing a validation experiment with both metabolomics and transcriptomics profiles.

As most species-specific reactions take place in peripheral pathways (Fig. 3), it is not immediately apparent that there would be species-specific differences in the ability to produce a metabolic biomarker under physiological constraints. However, TIMBR biomarker predictions can account for species-specific differences in gene expression patterns with an explicit mapping of species-specific GPR rules as accounted for in iRno and iHsa (Fig. 2). In addition, the TIMBR algorithm was designed to use relative expression changes, because absolute expression was difficult to compare across species-specific microarray technologies. To overcome this limitation, absolute expression from untargeted transcriptomics or proteomics experiments could be integrated into the TIMBR algorithm to further modify reaction weights. Alternatively, absolute expression can be used to constrain generic rat and human metabolic networks into tissue-specific models before simulating biomarker predictions. By disabling reactions in tissue-specific models, newly introduced species-specific differences could negatively impact comparative analyses between rat and biomarker predictions.

**Metabolic network visualization.** Metabolic network maps of reactions and metabolites from iRno and iHsa were generated using MetDraw[70] in the Python programming environment (http://www.python.org). An Systems Biology Markup Language (SBML) file containing the superset of reactions capable of carrying flux in either iRno or iHsa under relaxed physiological conditions was input into MetDraw for visualization of global networks. SBML files containing the subset of rat reactions with non-zero flux values from TIMBR simulations of urea production in response to caffeine.

**Cell culture.** Primary rat hepatocytes (Invitrogen, Carlsbad, CA) or human HepG2 cells (ATCC via the Tissue Core Facility at the University of Virginia) were thawed and plated in William's E Media (WEM) with appropriate thawing and plating supplements according to manufacturer's instructions. Mycoplasma testing was performed for HepG2 cells using the MycoAlert System (Lonza, Walkersville, MD). Hepatocytes were plated into a 12-well tissue culture plate at a density of $\sim 6 \times 10^5$ cells in each well resulting in 85–90% confluence. The day after plating, cells were exposed to 0 or $10\,\mu M$ of theophylline (reconstituted in WEM + 0.1% dimethyl sulfoxide) for 24 h. After exposure, supernatants from four biological replicates for each condition were removed and frozen before running metabolite assays. Spearman's rank correlation coefficients were calculated to determine relationships between theophylline concentration and assay results from supernatants. Data presented were obtained from one experiment with rat and human hepatocytes. Separate rat and human pilot experiments were each performed once before obtaining the comparative results.

**Metabolite assays.** Levels of glucose, glutamic acid and urate from supernatants were measured using AmplexRed-based assays (Invitrogen: A22189, A12221 and A22181) according to manufacturer's directions for the Qubit fluorometer. Urea levels were measured using a colorimetric assay kit (Biovision, K375-100) according to manufacturer's directions. Following the assays, the background level of each measured metabolite in fresh WEM or WEM with theophylline was averaged ($n = 2$) and subtracted from the individual respective supernatant samples. Endpoint concentrations were determined by adding back the average baseline concentration of fresh media without theophylline. Endpoint concentration changes between theophylline ($n = 4$) and control treated samples ($n = 4$) were determined for each metabolite using an unpaired two-sided Student's $t$-test. Statistical significance was determined independently for rat and human hepatocyte experiments by correcting for multiple hypothesis testing across the four metabolites and applying a FDR-adjusted $q$-value threshold of 0.05.

**Code availability.** To promote reproducibility of this research, computer code (MATLAB and R scripts) related to this study are available at www.github.com/csbl/ratcon1. Constraint-based methods were implemented within MATLAB and R/Bioconductor programming environments using the Constraint-based Reconstruction and Analysis (COBRA) toolbox (www.opencobra.github.io/cobratoolbox), the RAVEN toolbox (www.biomet-toolbox.org) and the Systems Biology Library for R package (sybil package). Gurobi 6.0.5 (www.gurobi.com), Mosek (www.mosek.com) and GLPK (glpkAPI package) were used for solving linear programming problems on a 64-bit desktop computer running Windows 10.

**Data availability.** SBML formatted files of iRno and iHsa are publically available for download at www.github.com/csbl/ratcon1. Spreadsheet formatted files of iRno and iHsa that can be imported into the COBRA toolbox and the RAVEN toolbox are also included as Supplementary Data. Consensus orthology annotations mapped onto HMR2 reactions, which are used to generate Supplementary Fig. 3, are provided in Supplementary Data 2. Updated annotation information for reactions and metabolites that are not usually stored in SBML, COBRA and RAVEN formatted files can be found in Supplementary Data 1 and 3. Data needed to simulate metabolic tasks using the RAVEN toolbox are provided in Supplementary Data 4. Raw gene expression data are available in ArrayExpress with the primary accession codes E-MTAB-797 and E-MTAB-798. Normalized gene expression data and differential expression analysis results can be reproduced using data described in

Supplementary Data 6 and R code available at www.github.com/csbl/ratcon1. TIMBR predictions depicted in Figs 5–7 are provided in Supplementary Data 7. Additional source data needed to reproduce figures can be obtained via the code available at www.github.com/csbl/ratcon1. All other data supporting the findings of this study are available within the article and its Supplementary Information files.

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

## Acknowledgements

We thank Michael Carter for network annotations and Jennifer A. Bartell, Arvind K. Chavali and Kevin M. D'Auria for discussions. Support for this project was provided by the United States Department of Defense (W81XWH-14-C-0054 to J.A.P.). The opinions and assertions contained herein are the private views of the authors and are not to be construed as official or as reflecting the views of the US Army or of the US Department of Defense. Citations of commercial organizations or trade names in this report do not constitute an official Department of the Army endorsement or approval of the products or services of these organizations. This paper has been approved for public release with unlimited distribution.

## Author contributions

E.M.B., A.W. and J.A.P. conceived the study. E.M.B. and Z.I.L. reconstructed the networks. E.M.B., K.D.R. and Z.I.L. performed the computational analyses of the reconstructions. B.V.D., K.D.R. and G.L.K. performed the experimental validation of model predictions. E.B. wrote the initial draft of the manuscript. E.M.B., K.D.R., B.V.D., G.L.K., P.Y., A.W. and J.A.P. edited and wrote the final manuscript.

## Additional information

**Competing financial interests:** The authors declare no competing financial interests.

