## [Peer review file · Nature Communications]

Reviewers' comments:

Reviewer #1 (Remarks to the Author):

A. The first genome-scale reconstruction of *Rattus norvegicus*

3 metabolism, iRno, and a significantly improved reconstruction of human metabolism, ipsa, are described. Gene expression changes in response to pharmaceutical compounds and environmental toxicants were used to derive biomarkers. For this purpose, a new algorithm was developed: Transcriptionally-Inferred Metabolic Biomarker Response (TIMBR) estimates the feasibility of producing a metabolite given changes in gene expression.

Overall this paper is a very nice example of the power of models to understand toxicological responses in general, and species differences specifically.

B. This enormous work is an important step forward in the modeling of metabolism of both species. Sharing all of this in the supplementary material makes this a rich resource for others to build on.

C. The approach is sound. However, the description in the article itself is very superficial, often simply referring to extensive manual curation. It might be useful to describe more extensively in the main text how the GENRE is built, how it differs from more basic metabolic models, why the available rat network is so much smaller than the human, etc. I think a longer discussion of their progress (and lack thereof) of understanding metabolic pathways at a global level (compared to our understanding of gene networks) might flesh out the paper.

The use of orthologs to extrapolate from human to rats is nicely done, but it wasn't clear from the materials and methods if there was any attempt to manually check accuracy of the approach.

I'm not sure the example of vitamin C in the presence or absence of GULO is much of a test-case for the computational abilities of these models, since it is a somewhat simplistic example; the other examples are much stronger and should probably be emphasized with the vitamin C example in passing.

D. Does not apply.

E. The findings described offer some face-validity. These tools and the novel TIMBRE approach will only in the future show their full reliability and usefulness.

F. The TIMBRE approach is very promising and certainly represents a more ambitious attempt to understand species-specific metabolism with models than in the past. That having been said, the model is built on microarray data using a somewhat un-stringent cut-off for differential expression; a lot of assumptions (i.e. that downregulated reactions contributed more than upregulated to the reaction weights) and the predictive power appears hit or miss and comparing to experimental literature values (and it is unclear what the range or consistency of the experimental literature values are). This is to be expected given that is a proof-of-principle rather than an attempt to model, in detail, any specific reaction, but it might be a good idea to mention the potential problems in this approach and how it could be improved going forward (with more accurate kinetics, more consistent biomarker measurements, etc).

G. References are adequate. Reference 1 seems odd (avocado oil supplement on cardiovascular risk markers to make a general statement that rat research is important).

H. Fine. A hint that the model is shared here would be good.

Reviewer #2 (Remarks to the Author):

Blais et al presented a genome-scale reconstruction (GENRE) of *Rattus norvegicus* metabolism, iRno, and an improved reconstruction of human metabolism, iHsa. The authors performed manual curation and compared the metabolic differences between the models for rat and human. The authors tested the capabilities of the models by predicting biomarkers for rat and human hepatocytes through integration of gene expression changes in response to 76 pharmaceutical compounds and environmental toxicants. They also provided literature-based evidence for antipyretic and anti-gout medicines and identified common metabolite biomarkers common to rats and humans.

The ambition and conclusions of the paper are fitted with the scope of the journal. However, the presentation of the results is sometimes poor and unclear, and overall the rationale for generation of GENREs for rat is not well justified. In order to show the differences in the identified biomarkers specific to rat and human hepatocytes, the authors should generate additional experimental data. The readers of Nature Communications would expect novel results and experimental validation to support their predictions. Otherwise, it will be well-suited for a specialized journal. Below are some concerns:

General Concerns:

1) The authors generated two GENREs for rats and humans and this reviewer was expecting to see major (significant) differences in the number of reactions, metabolites and associated genes included in both models. The differences in the number of the reactions (7 reactions) and genes (3 genes) incorporated into both GENREs were extremely small whereas the number of the metabolites were identical. Considering that the paper is focused on the differences in the metabolism of the rats and humans, clear metabolic differences should be presented.

Table 1 also should include the number of the genes included in each model.

2) Incorporation of 1103 manually reconciled GPR relationship rules in to HMR2 is definitely a significant improvement. Comparing the GPR rules in Recon1 and Recon2, how many of the rules are present only in iHsa? Are there any differences in the GPR rules between human and rats? What is the confidence for these GPR rules in different human tissues? These issues should be discussed in the manuscript.

3) The authors had an extensive literature search and found (small) differences in the number of enzymatic reactions between rats and humans. However how these differences affected the response of the hepatocytes to different pharmaceutical compounds and environmental toxicants? It should be clarified in the results section.

4) The authors compared the growth of rat and human hepatocytes. It is not clear how human and rat hepatocytes models are generated. What are the differences in the number of reactions carrying fluxes? Can the differences in growth be explained by the differences in the component of biomass equation?

5) The authors focused on the differences in the response of rat and human hepatocytes. They should experimentally validate one of the biomarker specific to rats and humans to demonstrate the predictive power of their reconstructed models.

6) The authors use FBA to predict the maximum specific growth rate for rat and human cells. However, the inputs for these simulations are not well specified. This information is essential for assessment of these values.

Reviewer #3 (Remarks to the Author):

COMMENTS TO THE AUTHORS

In this manuscript, the authors detail a comparative systems biology analysis between rat and human metabolism. The value of this analysis is to improve our understanding of rat models that are used for the study of human biology and medicine. The authors highlight specific examples which are faithfully captured by the models, namely that of vitamin C metabolism, gene expression changes in hepatocyte cells, and biomarker predictions.

The contribution also highlights the importance of reconstructing metabolic systems in other organisms and the need for reconciliation of systems biology models - an issue that is both timely and appropriate in this current state of systems science. The issue of lack of animal/organism specific highly curated reconstructions has been mentioned in several papers recently, and this study addresses this need; *Nat Biotech* 32: 447-452 (2014), *Current Opinion in Biotechnology* 34: 105-109 (2015), *Brief Bioinform*: 1-12 (2015). Perhaps this point should be brought out in the paper to strengthen the case for its timeliness.

The overall the manuscript was very well thought out and developed. Their algorithm for toxicogenomic analysis is interesting but just the amount of work they put into constructing their 2 reconstructions is very impressive by itself. This MS should be accepted for publication in *Nature Communications*.

MAJOR CONCERNS

Since the rat metabolome is studied in the context of human metabolism, is it surprising that 96% of all genes/reactions are shared? (line 90- zero species-specific reactions)

As one of the main messages of this article is to elucidate differences between rat-human biology in order to further translational impact of using rats as models for the translation of therapeutic strategies, the authors should make a clear statement in the discussion to clearly state how models such as these could be used in this degree. While the retrospective validation they provide is convincing, it would strengthen their claim to understand how such models could be used to discover new biology or therapeutic strategies.

MINOR CONCERNS

The authors should reference (in the main text) the major differences between *iHsa* and *Recon2* as a result of their reconstruction process. Currently, they only mention the number of increased reactions that have been added to their model.

Line 365 "specific-specific" typo

Since the contribution heavily weights on GPR relationships, a one-sentence expanded description of what this relationship is (for readers outside of computational systems biology) is necessary.

Main text, line 50: Insert 'the' between 'Overall, comparative' for clarity.

Main text, line 233: Doubling times from predicted max growth rates are 14.44 and 17.33 hours respectively, should probably be stated rather than averaged.

Main text, comparative biomarker predictions (Lines 308-315): The outcome from comparing human/rat TIMBR predictions seems underdeveloped, while the results may not be appropriate for the main text, it may merit further exploration in a supplementary discussion.

Main text, line 342: Have the authors looked at the metabolic basis (on a reaction/pathway level) for the difference between caffeine/theophylline response? This would make the claim of 'mechanistic insights' stronger.

Main text, line 756-757: How ere the subset of 16 compounds/biomarkers selected for this figure?

Supplementary information, line 76-77: For cases where there were more than 2 rat orthologs for a human gene, how were the 2 replacement orthologs selected?

Supplementary information, Formatting complex GPR rules for TIMBR: Does the GPR formulation selected (i.e. redundancies grouped by subunit rather than complex) have any impact on the TIMBR predictions?

Reviewer #1 (Remarks to the Author):

A. The first genome-scale reconstruction of *Rattus norvegicus* metabolism, *iRno*, and a significantly improved reconstruction of human metabolism, *ihpa*, are described. Gene expression changes in response to pharmaceutical compounds and environmental toxicants were used to derive biomarkers. For this purpose, a new algorithm was developed: Transcriptionally-Inferred Metabolic Biomarker Response (TIMBR) estimates the feasibility of producing a metabolite given changes in gene expression. Overall this paper is a very nice example of the power of models to understand toxicological responses in general, and species differences specifically.

Response: We thank Reviewer #1 for summarizing key outcomes of this study and for offering positive remarks related to the overall contribution of this work.

B. This enormous work is an important step forward in the modeling of metabolism of both species. Sharing all of this in the supplementary material makes this a rich resource for others to build on.

Response: Thank you for these supportive comments related to the impact of this work as a resource for the modeling community.

C. The approach is sound. However, the description in the article itself is very superficial, often simply referring to extensive manual curation. It might be useful to describe more extensively in the main text how the GENRE is built, how it differs from more basic metabolic models, why the available rat network is so much smaller than the human, etc.

Response: We now describe the model building process more extensively in the **Fig. 1** caption and in the main text on lines 66-84. We have also added text at lines 78-80 to highlight differences between *iRno* and previously published rat metabolic networks that were created to perform metabolic flux analysis within the scope of central metabolism.

“GENREs of *Rattus norvegicus* (*iRno*) and *Homo sapiens* (*iHsa*) metabolism were constructed in parallel as an expansion of the Human Metabolic Reaction 2.0 database¹⁵ (HMR2). To enable comparative systems analyses, we created a unified reaction database termed Ratcon1 that includes the superset of metabolic and transport reactions that occur in rats and humans. Each GENRE also includes gene-protein-reaction (GPR) rules to describe genotype to phenotype relationships that are organism-specific. To establish an initial draft GENRE of rat metabolism, we used orthology information to replace human GPR rules with rat GPR rules (**Fig. 1a**). Next, we updated *iRno* and *iHsa* in parallel with 169 new reactions, 1103 manually reconciled GPR relationship rules, and over 5000 additional references to experimental literature and annotation databases²⁸⁻³⁰ (**Fig. 1b**; **Supplementary Table 1**). Compared to previous human and mouse GENREs^{12,13,15,31,32}, *iRno* and *iHsa* captured the highest numbers of total reactions, enzymatic reactions, reactions associated with complex GPR rules, and annotations to external databases (**Table I**). Because previous rat metabolic networks were created for the purpose of metabolic flux

analysis within the scope of central metabolism, to our knowledge *iRno* represents the first genome-scale network model of rat metabolism. Furthermore, all reactions were reconciled for potential differences between rat and human networks which has not previously been described for existing mammalian networks (**Table I**). As a result, *iRno* and *iHsa* represent two of the most comprehensive metabolic reconstructions and the first pair of mammalian metabolic networks reconciled for comparative analyses to date.”

[66-84]

I think a longer discussion of their progress (and lack thereof) of understanding metabolic pathways at a global level (compared to our understanding of gene networks) might flesh out the paper.

Response: We have modified discussions on lines 427-436 to highlight progress on understanding metabolic pathways from a global perspective and included a statement to acknowledge the current understandings of gene networks on lines 435-436.

“With an improved understanding of rat and human metabolism at a global level, we can partially address inherent limitations in the use of rats to study human physiology and disease. Despite a surprisingly small number of species-specific differences at the genome level, individual differences at the gene level can alter network functionality. Unlike rats, humans exclusively rely on dietary sources of vitamin C, which may obfuscate the clinical translation of rat studies that have described vitamin C as a potential biomarker^{7,9}. The abundance and absence of β -muricholic acids in rats and humans, respectively, can have substantial implications within the context of toxicology because bile acids are frequently used as blood-based biomarkers of liver damage⁸. Furthermore, species-specific differences in gene network regulation and downstream cellular responses to stimuli have been observed but are not well understood⁶⁷.”

[427-436]

The use of orthologs to extrapolate from human to rats is nicely done, but it wasn't clear from the materials and methods if there was any attempt to manually check accuracy of the approach.

Response: We have added a statement on lines [498-505] to describe our efforts to check the accuracy of rat GPR rules as part of the curation process. Interestingly, the need to filter orthology annotations was only realized after manually checking rat GPR rules in the context of human GPR rules. As we continued to update GPR rules based on experimental literature and functional annotations, we were generally satisfied with the consensus-based approach.

“After applying automated methods to construct draft GENREs, rat and human GPR rules were manually examined in the context of experimental literature and functional databases. As part of the network reconciliation process, discrepancies were resolved for both rat and human GPR rules when possible. Manual changes frequently affected both rat and human GPR rules (**Supplementary Table 1**), highlighting the reliance of this automated approach on the accuracy of GPR rules from the original model and the quality of orthology annotations. Interestingly, the need to filter orthology annotations was only recognized after manually inspecting evidence for both rat and human GPR rules.”

[498-505]

I'm not sure the example of vitamin C in the presence or absence of GULO is much of a test-case for the computational abilities of these models, since it is a somewhat simplistic example; the other examples are much stronger and should probably be emphasized with the vitamin C example in passing.

Response: Thank you for this constructive feedback. We have modified the main text to deemphasize metabolic differences associated with vitamin C synthesis [154-161].

Unlike humans, rats are capable of producing vitamin C (ascorbate) (**Fig. 3a**) and are thus resistant to scurvy³⁹. *iRno* and *iHsa* captured this species-specific phenomenon with a new task that simulated *de novo* vitamin C synthesis in a glucose minimal media environment (**Supplementary Table 4**). The rat-specific enzyme, *Gulo*, is known to be responsible for this functional difference which limited the use of rat as a model organism for scurvy in the early 20th century^{39,40}. This species-specific difference provides a simplistic example of how *iRno* and *iHsa* can be leveraged to investigate genetic engineering strategies that bridge the gap between rat and human biology (see **Supplementary Fig. 4**).
[154-161]

D. Does not apply.

E. The findings described offer some face-validity. These tools and the novel TIMBRE approach will only in the future show their full reliability and usefulness.

Response: Thank you for this comment. We hope that the new experimental results related to TIMBR predictions described in response to point 5 of Reviewer #2's remarks provide additional confidence in these findings.

F. The TIMBRE approach is very promising and certainly represents a more ambitious attempt to understand species-specific metabolism with models than in the past. That having been said, the model is built on microarray data using a somewhat un-stringent cut-off for differential expression; a lot of assumptions (i.e. that downregulated reactions contributed more than upregulated to the reaction weights) and the predictive power appears hit or miss and comparing to experimental literature values (and it is unclear what the range or consistency of the experimental literature values are). This is to be expected given that is a proof-of-principle rather than an attempt to model, in detail, any specific reaction, but it might be a good idea to mention the potential problems in this approach and how it could be improved going forward (with more accurate kinetics, more consistent biomarker measurements, etc).

Response: We have added text [712-728] to discuss potential limitations and improvements of the bioinformatics approach and the TIMBR algorithm. We also modified descriptions of the TIMBR method [658-665] to clarify how one set of expression changes is transformed into two sets of reaction weights: reaction weights for the control condition are decreased by

downregulated genes and reaction weights for the treatment condition are decreased by upregulated genes. To clarify the range and consistency of values used for validation, we included averages, standard deviations, and p-values for individual metabolic changes to **Supplementary Table 7** for experimental values from literature as well as new data generated in response to Reviewer #2's remarks.

G. References are adequate. Reference 1 seems odd (avocado oil supplement on cardiovascular risk markers to make a general statement that rat research is important).

Response: We have replaced Reference 1 on line 19 with a review on systems toxicology, preclinical drug development and biomarker discovery using rats.

H. Fine. A hint that the model its shared here would be good.

Response: We have modified the text to indicate the availability of both the rat and human metabolic models online and within supplementary files. Code availability and Data availability statements have been added to include additional details related to sharing of these models (<http://bme.virginia.edu/csbl/downloads>) and relevant code (<http://github.com/csbl/ratcon1>).

Reviewer #2 (Remarks to the Author):

Blais et al presented a genome-scale reconstruction (GENRE) of *Rattus norvegicus* metabolism, iRno, and an improved reconstruction of human metabolism, iHsa. The authors performed manual curation and compared the metabolic differences between the models for rat and human. The authors tested the capabilities of the models by predicting biomarkers for rat and human hepatocytes through integration of gene expression changes in response to 76 pharmaceutical compounds and environmental toxicants. They also provided literature-based evidence for antipyretic and anti-gout medicines and identified common metabolite biomarkers common to rats and humans. The ambition and conclusions of the paper are fitted with the scope of the journal. However, the presentation of the results is sometimes poor and unclear, and overall the rationale for generation of GENREs for rat is not well justified. In order to show the differences in the identified biomarkers specific to rat and human hepatocytes, the authors should generate additional experimental data. The readers of Nature Communications would expect novel results and experimental validation to support their predictions. Otherwise, it will be well-suited for a specialized journal. Below are some concerns:

Response: We would like to thank Reviewer #2 for summarizing the key outcomes of this study and for providing constructive feedback to improve the impact and clarity of this work. We have modified the introduction of the main text to include additional justification for building a GENRE of rat metabolism, which incorporates suggestions by Reviewer #3 to discuss the need for animal metabolic networks models and the need for examples of comparative metabolic network analyses. We trust that the revised manuscript has addressed Reviewer #2's general concerns related to presentation of results and the need for additional experimental validation. The new experimental data and associated analyses are described in more detail below.

General Concerns:

1) The authors generated two GENREs for rats and humans and this reviewer was expecting to see major (significant) differences in the number of reactions, metabolites and associated genes included in both models. The differences in the number of the reactions (7 reactions) and genes (3 genes) incorporated into both GENREs were extremely small whereas the number of the metabolites were identical. Considering that the paper is focused on the differences in the metabolism of the rats and humans, clear metabolic differences should be presented.

Response: We also expected to find a large number of species-specific differences in the metabolic capabilities of rats and humans. Surprisingly, we were only able to identify 1 human gene and 7 rat genes that could be mapped to species-specific reactions with high confidence. We have updated the main text [102-111] to highlight why this result was unexpected and to clarify rat and human metabolic differences at the reaction-level. The total numbers of metabolites are identical across the two models because each metabolite can be consumed or produced by at least one other reaction. For example, exogenous nonhuman sialic acid can be used as a substrate by human sialyltransferases. While there were fewer differences in the total numbers of genes/reactions/metabolites than anticipated, there were significant differences in

the mappings between genes and reactions (**Fig. 2**) which can have significant impact on functional predictions of the network models.

“After network reconciliation, there was a high degree of confidence in the conserved metabolic functionality of *iRno* and *iHsa*. Unexpectedly, we found that rat and human metabolic networks were distinguished by as few as 8 unique enzymes. At the genome-scale, 99.6% of all gene-associated reactions were annotated with both rat and human genes (**Supplementary Table 3**). We simulated the effects of species-specific differences on network connectivity and found that 41 biochemical or transport reactions were uniquely capable of carrying flux in either *iRno* or *iHsa*. This result was unanticipated because 739 flux carrying reactions from HMR2 had been disabled in the draft GENRE of rat metabolism. Despite extensive efforts to identify metabolic activities unique to rat or human genomes (see **Supplementary Methods**), most metabolic subsystems included zero species-specific reactions after manual curation (**Fig. 1c**).”
[102-111]

Table 1 also should include the number of the genes included in each model.

Response: We modified **Table 1** to include the total numbers of genes represented in each model, replacing the numbers of model genes represented by GPR rules with fewer than 10 genes. Note that total numbers of genes do not necessarily reflect the comprehensiveness of metabolic network reconstructions, particularly when reactions like “protein kinase activity” are associated with highly redundant GPR rules. We also modified **Table 1** to include average GPR sizes across gene-associated reactions in each model.

2) Incorporation of 1103 manually reconciled GPR relationship rules in to HMR2 is definitely a significant improvement. Comparing the GPR rules in Recon1 and Recon2, how many of the rules are present only in *iHsa*?

Response: Of the 1103 reactions with manually reconciled GPR rules, 256 are unique to *iHsa* and not found in Recon 1, Recon2, or HMR2. However, direct GPR comparisons between models can be difficult to interpret, so we focused on comparisons between unique genes in the main text. We have modified the main text [244-254] to summarize important differences between *iHsa* and previous human metabolic networks:

“Manual curation introduced major differences between *iHsa* and previous human GENREs beyond species-specific pathways. The average GPR size across enzymatic reactions decreased from 3.89 in HMR2 to 2.97 in *iHsa* (**Table I**) after removing 1424 human genes, most of which were associated with signaling pathways (**Supplementary Table 1**). With an average GPR size of 1.97 (**Table I**), Recon 2 shared 1531 of its 1733 genes with *iHsa*. Although Recon 2 shared 1677 genes with HMR2, *iHsa* and Recon 2 included complex GPR relationships that were absent in HMR2 (**Table I**). In addition to modifying GPR rules, we also removed several reactions that were present in HMR2 (**Supplementary Table 1**). Unlike Recon 1, Recon 2, and HMR2, *iHsa* does not include thermodynamically infeasible reaction loops that drive unrealistic rates of ATP production and H₂O₂ detoxification with limited nutrients (see **Supplementary Methods**).”
[244-254]

Are there any differences in the GPR rules between human and rats?

Response: We state that GPR sizes were consistent for nearly 80% of reactions associated with both rat and human genes [128-130], so at least 20% of shared enzymatic reactions were different by 1 or more genes. An example of a GPR rule that is different between humans and rats is illustrated in **Fig. 2c**.

What is the confidence for these GPR rules in different human tissues? These issues should be discussed in the manuscript.

Response: We did not assign confidence to updated GPR rules in the context of different human tissues. Instead, we formulated GPR rules to capture all genotype to phenotype relationships possible. This allows users to instantiate tissue-specific models with the integration of expression data. Future studies that analyze tissue-specific metabolic networks based on *iRno* and *iHsa* would be necessary to evaluate the confidence of these GPR rules in tissue-specific contexts. We have updated the main text [134-140] to discuss tissue-specific models:

“Despite individual variations in GPR sizes and a handful of species-specific reactions, rat and human GPR rules remained relatively balanced at the genome-scale (**Fig. 2a**) and were not suggestive of any global differences in robustness within metabolism. However, these rat and human GPR formulations do not reflect potential tissue-specific differences in gene expression or enzyme regulation. Integration of such data into *iHsa* and *iRno* to generate tissue-specific models could be of significant interest in numerous biological contexts^{13-15,22,23}.”
[134-140]

3) The authors had an extensive literature search and found (small) differences in the number of enzymatic reactions between rats and humans. However how these differences affected the response of the hepatocytes to different pharmaceutical compounds and environmental toxicants? It should be clarified in the results section.

Response: We have updated the main text on lines 322-327 to discuss the role of species-specific reactions in making the comparative biomarker predictions:

“Since most species-specific reactions take place in peripheral pathways (**Fig. 3**), it is not immediately apparent that there would be species-specific differences in the ability to produce a metabolic biomarker under physiological constraints. However, TIMBR biomarker predictions can account for differences in gene expression patterns, which represent species-specific responses to a compound, with an explicit mapping of species-specific GPR rules as accounted for in *iRno* and *iHsa* (**Fig. 2**).”
[322-327]

4) The authors compared the growth of rat and human hepatocytes. It is not clear how human and rat hepatocytes models are generated. What are the differences in the number of reactions carrying fluxes? Can the differences in growth be explained by the differences in the component of biomass equation?

Response: Thank you for these questions related to reconciled biomass formulations for rat and human hepatocytes. Growth rate differences are attributed to species-specific differences in the biomass compositions, specifically minor differences in the amount of essential amino acids in one unit of biomass. We have updated the methods [575-581] to reflect this observation and to clarify that physiological constraints for exchange reactions based on previously reported uptake and production rates for hepatocytes were the only hepatocyte-specific constraints applied to the generic models for growth rate simulations and biomarker predictions.

“Strict physiological constraints were defined as inputs (**Supplementary Fig. 7**) for quantitative predictions of maximum growth rates using the unified biomass reaction as the cellular objective for *iRno* and *iHsa* (see **Supplementary Methods** for additional details). It is important to note that upper and lower bound constraints were consistent across all shared rat and human reactions, including physiological constraints for exchange reactions. Thus, species-specific differences in growth rate predictions could be explained by minor differences in biomass requirements for essential amino acids.”
[575-581]

5) The authors focused on the differences in the response of rat and human hepatocytes. They should experimentally validate one of the biomarker specific to rats and humans to demonstrate the predictive power of their reconstructed models.

Response: We have generated additional experimental data from primary rat hepatocytes for which there were human data available to validate biomarker predictions. We have modified the main text to include additional methods [747-773], results [386-414], and discussions [439-445] related to these new experiments. The experiments were designed to test claims that appeared in the original manuscript, specifically those related to human-specific predictions for theophylline (**Fig. 6a**) and similar rat predictions for theophylline and caffeine (**Fig. 6d**). In total, we validated theophylline-induced biomarker changes predicted by *iRno* for five metabolites: glucose, urate, urea, citrulline, and glutamate. We present results from these new experiments in **Fig. 7**. We also updated **Fig. 6d-e** to highlight all five metabolites with new experimental results (urea and urate were originally shown).

To demonstrate the power of *iRno* and *iHsa* in predicting species-specific differences, we experimentally measured theophylline-induced changes in glucose and urate levels for rat hepatocytes. Rat-specific predictions and compared the results to literature-based evidence in humans that supported human-specific predictions.

Because rat biomarker predictions for theophylline and caffeine were strongly correlated, we also measured theophylline-induced changes for three metabolites (glutamate, urea, and citrulline) that could be compared with caffeine-induced changes shown in **Figure 5**. Results

from these experiments demonstrated the power of these models in predicting changes in metabolite levels using comparative toxicogenomics data.

6) The authors use FBA to predict the maximum specific growth rate for rat and human cells. However, the inputs for these simulations are not well specified. This information is essential for assessment of these values.

Response: We have updated the main text on lines [574-581] to define model inputs (physiological constraints for exchange reactions), model outputs (biomass compositions based on various independent datasets), and model assumptions (steady-state fluxes) for specific growth rate predictions. In-depth descriptions of how these parameters were formulated are available as **Supplementary Information**.

Reviewer #3 (Remarks to the Author):

In this manuscript, the authors detail a comparative systems biology analysis between rat and human metabolism. The value of this analysis is to improve our understanding of rat models that are used for the study of human biology and medicine. The authors highlight specific examples which are faithfully captured by the models, namely that of vitamin C metabolism, gene expression changes in hepatocyte cells, and biomarker predictions.

Response: We would like to thank Reviewer #3 for these remarks that emphasize the value of these comparative analyses in understanding rat models of human biology and medicine.

The contribution also highlights the importance of reconstructing metabolic systems in other organisms and the need for reconciliation of systems biology models- an issue that is both timely and appropriate in this current state of systems science. The issue of lack of animal/organism specific highly curated reconstructions has been mentioned in several papers recently, and this study addresses this need; Nat Biotech 32: 447-452 (2014), Current Opinion in Biotechnology 34: 105-109 (2015), Brief Bioinform: 1-12 (2015). Perhaps this point should be brought out in the paper to strengthen the case for its timeliness.

Response: Thank you for recommending these articles, they are highly relevant to this work. We have modified the introduction of the main text [28-48] to emphasize the need for high quality animal metabolic networks and the importance of network reconciliation, especially for model organisms of human diseases. Two of these were added as #18 and #27.

“A GENome-scale Network REconstruction (GENRE) of metabolism acts as a repository for all known biochemical and transport reactions for an organism. Several GENREs with thousands of human genes have been published¹²⁻¹⁵ while only core metabolic networks with dozens of genes are available for rat^{16,17}. A high-quality GENRE of rat metabolism is needed to bridge the knowledge gap that exists between humans and rats in clinical and basic science applications. The limited availability of highly curated GENREs for rats and other animals has been attributed to the substantial efforts required to manually construct a GENRE based on information from biochemical databases, genome annotations, and literature evidence¹⁸.

A comprehensive collection of metabolic differences between rats and humans would be a valuable resource for understanding the applicability as well as the limitations of rats in preclinical drug development and biomarker discovery^{10,11}. Human GENREs have been used to predict metabolic biomarkers for inborn errors of metabolism (IEMs)^{13,19} and to analyze the metabolic effects of therapeutic strategies in the context of cancers, toxicology, and diabetes^{14,15,20,21}.

Computational methods for integrating gene and protein expression measurements into GENREs have been developed to generate context-specific metabolic networks and enable comparative predictions across individual patients, treatment conditions, and tissue-types^{15,22-24}. Resolving metabolic differences between rat and human GENREs would enable cross-species comparisons as previously described for bacterial GENREs^{25,26}. However, the lack of unified standards for metabolic networks²⁷ has limited the development of computational frameworks for analyzing species-specific differences between GENREs.”

[28-48]

The overall the manuscript was very well thought out and developed. Their algorithm for toxicogenomic analysis is interesting but just the amount of work they put into constructing their 2 reconstructions is very impressive by itself. This MS should be accepted for publication in Nature Communications.

Response: We appreciate the encouragement.

MAJOR CONCERNS

Since the rat metabolome is studied in the context of human metabolism, is it surprising that 96% of all genes/reactions are shared? (line 90- zero species-specific reactions)

Response: Yes! We were very surprised to find that the majority of reactions were shared by both rats and humans. We originally anticipated that we could easily find at least 10 human-specific reactions in addition to more than 50 reactions originally annotated as human-specific after automated generation of the initial draft rat reconstruction. Surprisingly, we found evidence of rat orthology for all but 1 of the reactions originally annotated as human-specific and we did not identify any new human-specific annotations during the manual curation process. We have updated the main text on lines [102-111] to convey our findings related to species-specific differences, facilitating the transition to the analysis shown in **Figure 1c**. Because species-specific differences rely on evidence for the existence of function in one organism and evidence for the absence of function in another, species-specific differences may remain undiscovered due to insufficient evidence related to rat metabolism, particularly for subsystems in which rat genes are studied less extensively relative to human genes.

“After network reconciliation, there was a high degree of confidence in the conserved metabolic functionality of *iRno* and *iHsa*. Unexpectedly, we found that rat and human metabolic networks were distinguished by as few as 8 unique enzymes. At the genome-scale, 99.6% of all gene-associated reactions were annotated with both rat and human genes (**Supplementary Table 3**). We simulated the effects of species-specific differences on network connectivity and found that 41 biochemical or transport reactions were uniquely capable of carrying flux in either *iRno* or *iHsa*. This result was unanticipated because 739 flux carrying reactions from HMR2 had been disabled in the draft GENRE of rat metabolism. Despite extensive efforts to identify metabolic activities unique to rat or human genomes (see **Supplementary Methods**), most metabolic subsystems included zero species-specific reactions after manual curation (**Fig. 1c**).”

[102-111]

As one of the main messages of this article is to elucidate differences between rat-human biology in order to further translational impact of using rats as models for the translation of therapeutic strategies, the authors should make a clear statement in the discussion to clearly state how models such as these could be used in this degree. While the retrospective validation they provide is convincing, it would strengthen their claim to understand how such models could be used to discover new biology or therapeutic strategies.

Response: We have expanded discussions in the main text [439-445] to state how these reconciled rat and human metabolic networks can be used to translate therapeutic strategies from preclinical studies to clinical trials.

“The comparative toxicogenomics workflow developed in this study could be used to further the translational impact of rats in biomarker discovery by highlighting metabolic biomarkers that should be avoided. Using relative changes in gene expression, we demonstrated the sensitivity of the TIMBR algorithm in predicting species-specific differences related to glucose and urate production in response to theophylline. With the ability to analyze biomarker predictions between treatments and across metabolites, TIMBR could also be informative in prioritizing biomarkers that are sensitive for a specific toxicological response.”
[439-445]

MINOR CONCERNS

The authors should reference (in the main text) the major differences between *iHsa* and Recon2 as a result of their reconstruction process. Currently, they only mention the number of increased reactions that have been added to their model.

Response: We have added descriptions of major differences between *iHsa* and Recon 2 in the main text on lines [231-242] related to network reconciliation and on lines [244-254] related to other curation efforts. In the article describing HMR2, Mardinoglu *et al.* stated that all reactions and genes from Recon 2 were also present in HMR2. Although we could not independently confirm this claim without annotations between all Recon 2 and HMR2 reaction identifiers, we have manually verified that newly added reactions were mostly absent from Recon 2.

“Network reconciliation significantly improved both *iRno* and *iHsa*. Although most species-specific functions described above are unique to rats, we discovered that previous human GENREs included rat-specific reactions associated with purine degradation, nonhuman sialic acid synthesis, and glycine metabolism. As a result, rat-specific reactions were not only added to *iRno* but also removed from *iHsa*. By resolving species-specific differences in the purine degradation pathway, we removed reactions from *iRno* and *iHsa* that allowed Recon 2¹³ and HMR2¹⁵ to degrade urate into urea (**Supplementary Table 1**), a function known to be absent in mammals but present in other vertebrates including fish⁴². While curating bile acid metabolism, we removed intracellular reactions involved in secondary bile acid synthesis (**Fig. 3f**) that should only take place in the mammalian gut and added new bile acid transport reactions (**Supplementary Table 1**). These examples highlight how reconciling differences between rat and human metabolism guided the improvement of *iHsa* compared to HMR2 and Recon 2.

Manual curation introduced major differences between *iHsa* and previous human GENREs beyond species-specific pathways. The average GPR size across enzymatic reactions decreased from 3.89 in HMR2 to 2.97 in *iHsa* (**Table I**) after removing 1424 human genes, most of which were associated with signaling pathways (**Supplementary Table 1**). With an average GPR size of 1.97 (**Table I**), Recon 2 shared 1531 of its 1733 genes with *iHsa*. Although Recon 2 shared 1677 genes with HMR2, *iHsa* and Recon 2 included complex GPR relationships that were absent in HMR2 (**Table I**). In addition to modifying GPR rules, we also removed several reactions that were present in HMR2 (**Supplementary Table 1**). Unlike Recon 1, Recon 2, and HMR2, *iHsa* does not include thermodynamically infeasible reaction loops that drive unrealistic rates of ATP production and H₂O₂ detoxification with limited nutrients (see **Supplementary Methods**).” [231-254]

Line 365 "specific-specific" typo

Response: Fixed on line 425, thank you.

Since the contribution heavily weights on GPR relationships, a one-sentence expanded description of what this relationship is (for readers outside of computational systems biology) is necessary.

Response: We have added text [69-71] to describe GPR relationships more appropriately for a broader audience.

“GENREs of *Rattus norvegicus* (*iRno*) and *Homo sapiens* (*iHsa*) metabolism were constructed in parallel as an expansion of the Human Metabolic Reaction 2.0 database¹⁵ (HMR2). To enable comparative systems analyses, we created a unified reaction database termed Ratcon1 that includes the superset of metabolic and transport reactions that occur in rats and humans. Each GENRE also includes gene-protein-reaction (GPR) rules to describe genotype to phenotype relationships that are organism-specific. To establish an initial draft GENRE of rat metabolism, we used orthology information to replace human GPR rules with rat GPR rules (**Fig. 1a**).” [66-72]

Main text, line 50: Insert 'the' between 'Overall, comparative' for clarity.

Response: Fixed on line 60, thank you.

Main text, line 233: Doubling times from predicted max growth rates are 14.44 and 17.33 hours respectively, should probably be stated rather than averaged.

Response: Fixed on line 276, thank you. Relatedly, we have included text on lines [575-581] to address Reviewer #2's question about the underlying reason rat and human growth rate predictions were different.

Main text, comparative biomarker predictions (Lines 308-315): The outcome from comparing human/rat TIMBR predictions seems underdeveloped, while the results may not be appropriate for the main text, it may merit further exploration in a supplementary discussion.

Response: Thank you for this comment. We have further developed the comparative biomarker predictions for theophylline with additional experimental validation, as suggested by Reviewer #2 and the Editor (see above). We also explored the mechanistic rationale behind some of the predictions, as described below.

Main text, line 342: Have the authors looked at the metabolic basis (on a reaction/pathway level) for the difference between caffeine/theophylline response? This would make the claim of 'mechanistic insights' stronger.

Response: To address this comment, we have investigated urate biomarker predictions for caffeine and theophylline and included additional analyses [391-394] in the newly added **Supplementary Fig. 8**.

“By comparing reaction weights and fluxes associated with urate production, we found that shared reactions involved in purine synthesis and purine degradation were uniquely upregulated in human hepatocytes by theophylline and not by caffeine (**Supplementary Fig. 8**).”
[391-394]

Main text, line 756-757: How are the subset of 16 compounds/biomarkers selected for this figure?

Response: We expanded the main text on lines [707-711] to include rationale for compound/biomarker selections: The treatments selected for this heatmap were chosen to include antipyretic compounds, xanthine oxidase inhibitors, and xanthine derivatives. Potential biomarkers were manually selected to include metabolites with validation data in Figure 5, PGE2 and urate which are directly downstream of enzymes targeted by cyclooxygenase inhibitors (antipyretics) and xanthine oxidase inhibitors, and a handful of common metabolites.

Supplementary information, line 76-77: For cases where there were more than 2 rat orthologs for a human gene, how were the 2 replacement orthologs selected?

Response: We have expanded the supplementary text starting at line 234 to describe how rat orthologs for individual human genes based on a variety of metrics. R scripts necessary to calculate database counts for ortholog pairs and orthology ranks for replacement genes are available on the GitHub repository.

Supplementary information, Formatting complex GPR rules for TIMBR: Does the GPR formulation selected (i.e. redundancies grouped by subunit rather than complex) have any impact on the TIMBR predictions?

Response: By grouping redundancies together [A & (B or C)], TIMBR reaction weights represent the subunit with the largest log fold change after averaging across redundancies independently for each subunit: $\text{largest}(\text{mean}(A), \text{mean}(B,C))$. By applying a similar approach with the alternative GPR rule format (A & B) or (A & C), TIMBR reaction weights represent an average of the largest log fold changes within all possible protein complexes: $\text{mean}(\text{largest}(A,B), \text{largest}(A,C))$. Changing the order of these operations would only have a minor impact on TIMBR predictions; however, we found the biological justifications for treating subunits independently more meaningful than the alternatives.

Reviewers' comments:

Reviewer #1 (Remarks to the Author):

The authors have addressed all my comments in an adequate manner.

Reviewer #2 (Remarks to the Author):

This manuscript mainly focuses on the differences between the metabolism of rat and human, and the response of the rat and human hepatocytes to different pharmaceutical compounds and environmental toxicants. In the previous version of the manuscript, I asked the authors to generate experimental data to show the differences in the identified biomarkers in human and rat hepatocytes when they are treated with a specific compound. The authors only showed the response of the rat hepatocytes to theophylline and did not generate any experimental data for human hepatocytes. The ideal experimental design should include a specific compound and its different effect on human and rat hepatocytes. The results of the experiments should also be compared with the predictions of the models.

I am not also convinced with the experimental results presented in Figure 7 of the paper. The level of citrullene did not increase in two of rat hepatocytes and the level of urate did not decrease in three of the rat hepatocytes as predicted by the rat model. Moreover, the level of the glucose decreased in all samples whereas the model predicted no differences.

In summary, the authors should demonstrate the value of building a rat and human specific model by testing the effect of a specific compound in human and rat hepatocytes and compare with the species specific model predictions.

Reviewer #3 (Remarks to the Author):

I find the revised version of this paper acceptable for publication.

Reviewer #2 Remarks:

This manuscript mainly focuses on the differences between the metabolism of rat and human, and the response of the rat and human hepatocytes to different pharmaceutical compounds and environmental toxicants. In the previous version of the manuscript, I asked the authors to generate experimental data to show the differences in the identified biomarkers in human and rat hepatocytes when they are treated with a specific compound. The authors only showed the response of the rat hepatocytes to theophylline and did not generate any experimental data for human hepatocytes. The ideal experimental design should include a specific compound and its different effect on human and rat hepatocytes.

Response: As recommended, we have conducted additional experiments in both human and rat hepatocytes to validate species-specific differences identified within our rat and human biomarker predictions. We re-designed our previous experiments to test the effects of theophylline on extracellular concentrations of glutamate, urate, glucose, and urea in cell cultures containing human hepatocytes (HepG2 cell line) or primary rat hepatocytes. As a result, we experimentally confirmed species-specific differences for all three metabolites tested with discordant rat and human predictions: glutamate (up in human, down in rat), urate (up in human, down in rat), and glucose (up in human, unchanged in rat). These new experimental results are now presented in **Figure 7** and provide more rigorous validation of our comparative predictions and higher quality evidence to support our rat and human networks.

The results of the experiments should also be compared with the predictions of the models.

Response: In addition to confirming species-specific differences between rat and human metabolism with these experiments, we have also included direct comparisons between our new experimental results and our original model predictions in both qualitative and quantitative fashions:

- Qualitatively, we compared the directionality of each prediction with the corresponding directionality of each experimental result the three categories: significantly elevated, significantly reduced, or insignificantly unchanged. As a result, 7 out of 8 predictions were in agreement while 1 was insignificantly upregulated but in the same direction as our prediction (see **Figure 7**).
- Quantitatively, we compared TIMBR production scores with average \log_2 fold changes across all 4 measured metabolites in **Figure 7e**. Using the same approach for validating caffeine-induced biomarker predictions in **Figure 5**, we found that theophylline-induced experimental changes were significantly correlated with our predicted changes in extracellular concentrations.

I am not also convinced with the experimental results presented in Figure 7 of the paper. The level of citrulline did not increase in two of rat hepatocytes and the level of urate did not decrease in three of the rat hepatocytes as predicted by the rat model. Moreover, the level of the glucose decreased in all samples whereas the model predicted no differences.

Response: With the feedback we received, we completely re-designed the experimental validation efforts as reported in this revised paper. The new data from our re-designed experiments are now presented in **Figure 7** instead with substantially improved outcomes primarily due to decreased experimental variability within each experimental group. We also simplified the interpretation of our statistical analyses (t-tests instead of Spearman's correlations) by skipping the previously tested low

dose of theophylline (2 uM) in our re-designed experiments. We attempted to include citrulline levels in our results but encountered unresolvable issues associated with detection limits of the assay.

In summary, the authors should demonstrate the value of building a rat and human specific model by testing the effect of a specific compound in human and rat hepatocytes and compare with the species specific model predictions.

Response: Thank you for these recommendations. Our new experimental results now demonstrate the value of *iRno* and *iHsa* with validated predictions for 3 species-specific effects of theophylline on rat and human hepatocyte metabolism. These validating experiments also add substantial value to this computational framework and thus increase the impact of this work because the TIMBR algorithm can integrate gene expression changes in response to any physiological perturbation. Overall, these new data and the accompanying revisions to the manuscript substantially improve the clarity of the results section and strengthen the quality of our experimental support for making comparative biomarker predictions.

Figure 7 – Validation of *iRno* and *iHsa* biomarker predictions in response to theophylline.

(a-d) Comparative biomarker predictions were experimentally validated *in vitro* using primary rat hepatocytes or an immortalized human hepatocyte cell line (HepG2). Extracellular changes in metabolite concentrations were measured after 24 hours of treatment with theophylline (10 µM) or control (0 µM). To the right of each metabolite, predictions are summarized as elevated (up arrow), reduced (down arrow), or unchanged (equals sign). Below each measured metabolite, FDR-adjusted q-values are displayed for rat and human experimental comparisons between treatment (n = 4) and control (n = 4) sample concentrations using an unpaired two-sided student's t-test. Individual points represent biological replicates from one rat and one human experiment. Of the 8 predictions tested, 7 were experimentally confirmed while urea from HepG2 cells (d) was insignificant but trending in the expected direction. Horizontal lines represent average metabolite concentrations from 2 fresh media replicates and indicate that all metabolites were being produced on average with the exception of glucose in HepG2 cells.

(e) Quantitative comparisons between model predictions and experimental results across metabolites. Positive and negative values represent elevated and reduced biomarkers based on TIMBR production scores (x-axis) and average \log_2 fold changes between treatment and control (y-axis) concentrations displayed in (a-d).

REVIEWERS' COMMENTS:

Reviewer #2 (Remarks to the Author):

I am satisfied with the revision. The new experiments are really valuable addition and the authors are to be congratulated for adding these. Very nice paper.